# $^{125}$Te and $^{57}$Fe nuclear resonance vibrational spectroscopic characterization of intermediate spin state mixed-valent dimers

Aleksa Radović [1], Justin T. Henthorn [1,7], Hongxin Wang [2], Deepak Prajapat [3], Ilya Sergeev [3], Nobumoto Nagasawa [4], Yoshitaka Yoda [4], Stephen P. Cramer [2] & George E. Cutsail III [1,5,6] ✉

Iron-sulfur clusters fulfill numerous roles throughout biology. The reduced [2Fe-2S]$^+$ cluster offers unique electronic and magnetic properties due to its mixed-valent nature and can serve as an essential model for understanding electron transfer, electron delocalization, and accessible spin states not only in mixed-valent dimers, but potentially larger iron sulfur clusters. Recently a series of mixed-valent diiron dichalcogenide complexes [L$_2$Fe$_2$Q$_2$]$^-$ (Q = S (**1**), Se (**2**), Te (**3**), L = 2,6-diisopropylphenyl β-diketiminate ligand) were synthesized and characterized, where complex **1** showed a typical $S = 1/2$ spin state, while complexes **2** and **3** exhibited intermediate $S = 3/2$ spin states, potentially enabled by the minimization of vibronic coupling. Here we studied the vibrational dynamics of the Fe and Te centers in these complexes using $^{57}$Fe and $^{125}$Te nuclear resonance vibrational spectroscopy (NRVS), coupled with DFT calculations. The findings suggest that heavy character of larger chalcogen atoms results in decreased vibronic coupling. The observation of an intermediate spin state is shown to be unattainable for lighter Fe$_2$Q$_2$ cores. This highlights the crucial role of vibronic coupling in modulating the electronic structure of mixed-valence systems and should enhance understanding of the electronic structure in more complex biological Fe-S clusters.

The [2Fe-2S] motif represents the simplest form of Fe-S clusters and is a very common biological prosthetic group responsible for many important functions in living organisms, including electron transfer reactions. In electron transfer processes, these clusters switch between oxidized, diferric [Fe$_2$S$_2$]$^{2+}$ and reduced, mixed-valent [Fe$_2$S$_2$]$^+$, forms[1,2]. The majority of biological and biomimetic mixed-valent [2Fe-2S]$^+$ clusters exhibit a $S_{tot} = 1/2$ ground spin state, a result of the antiferromagnetic coupling of the high-spin Fe$^{2+}$ ($S = 2$) and Fe$^{3+}$ ($S = 5/2$) ions[3–7]. These [2Fe-2S] clusters can also be viewed as building blocks of larger Fe-S clusters, including [4Fe-4S] and other more complex

biologically active sites such as the [7Fe9SMoC] cluster in many nitrogenases[8–10]. For higher nuclearity mixed-valent [4Fe-4S]$^+$ clusters, various intermediate spin states ($S_{tot} = 3/2$, and 5/2) and clear Fe$^{2.5+}$-Fe$^{2.5+}$ mixed-valent iron pairs are observed by Mössbauer spectroscopy and cannot be described by Heisenberg exchange coupling alone[11]. These more complex systems motivate the study and understanding of their electronic structure, particularly at the fundamental [2Fe-2S] unit level. In addition to the direct study of Fe-S clusters, it has been recently demonstrated that heavier chalcogenides, both Se and Te, can be introduced in biological and biomimetic Fe-S clusters as additional

[1]Max Planck Institute for Chemical Energy Conversion, Mülheim an der Ruhr, Germany. [2]SETI Institute, Mountain View, CA, USA. [3]Deutsches Elektronen-Synchrotron DESY, Hamburg, Germany. [4]Precision Spectroscopy Division, Sayo, Hyogo, Japan. [5]Institute of Inorganic Chemistry, University of Duisburg-Essen, Essen, Germany. [6]Department of Chemistry, Ludwig-Maximilians-Universität München, Munich, Germany. [7]Present address: School of Chemistry, University College Dublin, Dublin, Ireland. ✉e-mail: george.cutsail@cec.mpg.de

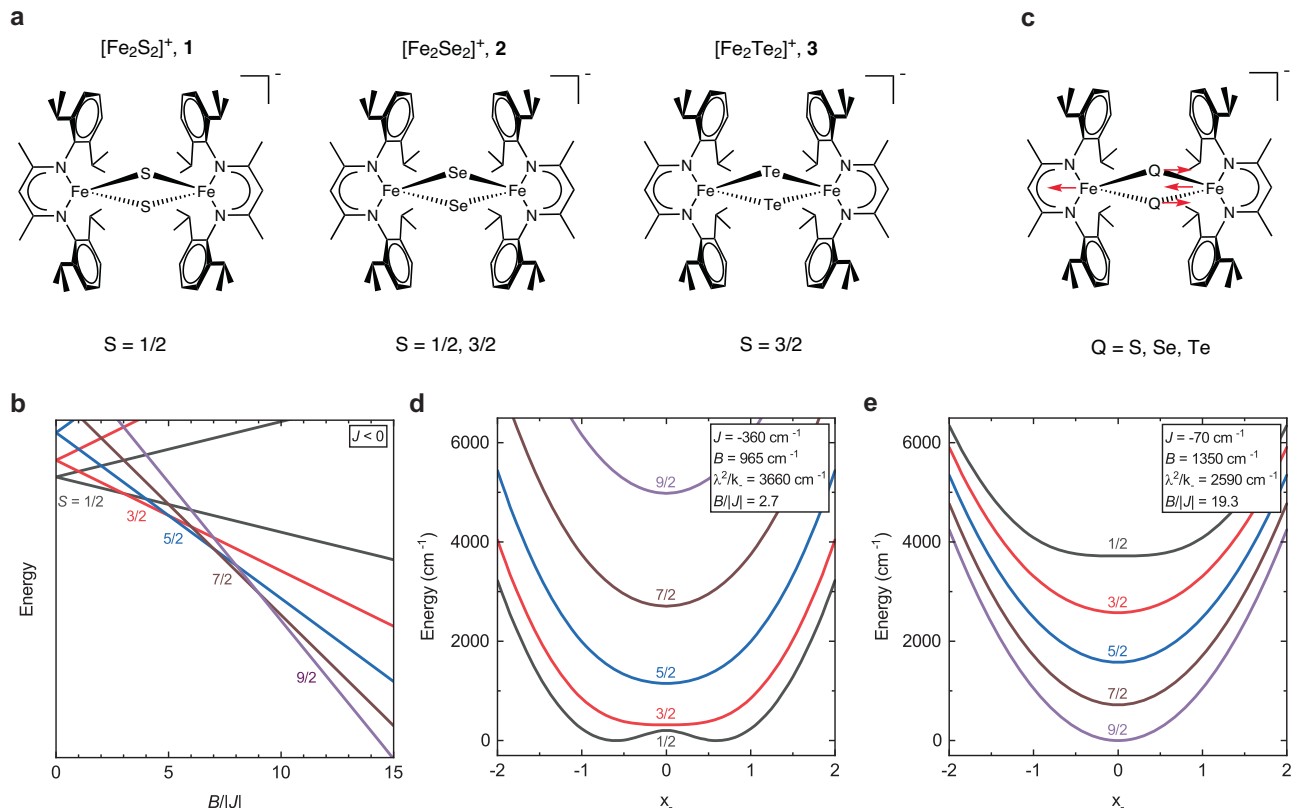

**Fig. 1 | Structures of studied complexes and example of potential energy surfaces. a** Structures of previously synthesized mixed-valent diiron dichalcogenide clusters. **b** Energy levels determined by Eq. 1, as a function of $B/|J|$ ratio, for antiferromagnetically coupled systems ($J < 0$). **c** Representation of PKS vibration for studied complexes. **d, e** Energy levels determined by Eq. 2 in the PKS coordinate for **d** $S = 1/2$ localized $[Fe_2S_2]^+$ complex and **e** $S = 9/2$ delocalized $[Fe_2(OH)_3(tmtacn)_2]^{2+}$ [23,24].

spectroscopic probes, offering additional site-specificity[12–16]. Replacement of S with heavier chalcogenides has been used to tune properties of various systems[17–19] including proteins[20–22], where they have an impact on electronic structure, thus it is important to evaluate the effect of chalcogen substitution on the electronic structure of biological and biomimetic clusters.

Previously, we reported on the synthesis and characterization of a series of mixed-valent diiron dichalcogenide clusters $[Fe_2Q_2]^+$ (Q = S (**1**), Se (**2**), Te (**3**)), supported by bulky β-diketiminate ligands (Fig. 1a)[23]. The $[Fe_2S_2]^+$ cluster exhibits a typical $S = 1/2$ ground spin state, while substitutions with selenide and telluride into the cluster significantly perturb the electronic structure and yield clear $S = 3/2$ spin state signatures that are well evidenced by electron paramagnetic resonance (EPR) spectroscopy. The EPR spectrum of the $[Fe_2Se_2]^+$ cluster at 4 K evidenced a physical mixture of $S = 1/2$ and $3/2$ spin states, while the $[Fe_2Te_2]^+$ cluster demonstrated an isolated $S = 3/2$ intermediate spin state[23]. The formalism of Heisenberg exchange coupling, $H = -2JS \cdot S$, is only able to rationalize the total high- (ferromagnetically coupled, $J > 0$) and low-spin (antiferromagnetically coupled $J < 0$) solutions, $S_{tot}$, for mixed-valent dimers: $S_{tot} = 9/2$ or $S_{tot} = 1/2$, respectively. The introduction of double exchange coupling, $B$, to the single unpaired electron lifts the degeneracy of the Heisenberg spin states to yield the energy eigenvalues given by the following equation:

$$E_\pm = -JS(S+1) \pm B(S+1/2) \qquad (1)$$

Uniquely, at high ratios of $B/|J|$ for antiferromagnetically coupled mixed-valent dimers, the total high-spin solution can be observed ($B/|J| > 9$, Fig. 1b).

Indeed, this formalism has been used to explain the $S = 9/2$ ground spin states of mixed-valent hydroxo-bridged iron dimers[24–28] as well as in a ferredoxin $[2Fe-2S]^+$ cluster[29–33]. The application of double exchange has not been limited to molecular iron dimers but also explains the high-spin ground states in other mixed-valent transition metal systems, including vanadium dimers[34,35]. In addition to Heisenberg and double exchange couplings, vibronic coupling can also have a significant influence on the cluster spin states by favoring electron localization and thus disfavoring the stabilization of intermediate spin states. First identified by Piepho, Krausz, and Schatz (PKS), the primary vibronic coupling mode, which favors electronic localization through desymmetrization of mixed-valent dimers, is a low-energy out-of-phase breathing mode (referred throughout here as the 'PKS vibration', Fig. 1c)[36,37]. The described Heisenberg double exchange model (Eq. 1) can be extended to include the influence of vibronic coupling by associating the electron-localizing PKS vibrational mode as a major vibronic contribution. The extended equation below (Eq. 2) is used to calculate the energy levels as a function of the vibronic coordinate (the PKS normal mode)[24,38,39].

$$E_\pm = -JS(S+1) + \frac{1}{2}\left(\frac{\lambda^2}{k_-}\right)x_-^2 \pm \left[\frac{1}{2}\left(\frac{\lambda^2}{k_-}\right)^2 x_-^2 + B^2\left(S+\frac{1}{2}\right)^2\right]^{1/2} \qquad (2)$$

In this equation, $(\lambda^2/k_-)$ is the vibronic coupling term (in cm⁻¹). For this term, $\lambda$ is calculated by $\lambda = k_-(\Delta Q_-) = k_-(n^{1/2}\Delta r)$, where $\Delta r$ is the difference in metal-ligand bond lengths between oxidized and reduced monomeric subunits–an approximation for nuclear displacements in PKS normal mode, and $n$ is the coordination number of each ion. Also in Eq. 2, $x_-$ is the dimensionless coordinate associated with normal

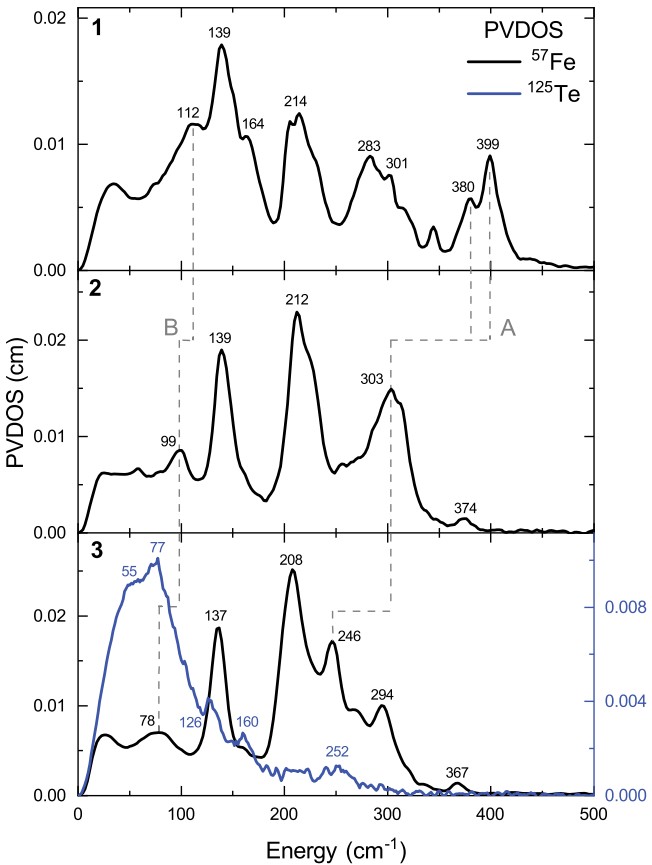

**Fig. 2 | Experimental $^{57}$Fe and $^{125}$Te nuclear resonance vibrational spectroscopy (NRVS) partial vibrational density of states (PVDOS).** $^{57}$Fe (black line) and $^{125}$Te (blue line) NRVS PVDOS spectra of complexes **1**–**3**. Gray dashed lines depict the shifting of the Fe-Q stretching (**A**) and lower energy core (**B**) vibrations.

mode in which $\Delta Q_-$ is the vibrational normal coordinate, $v_-$ is vibrational frequency, and $k_- = 4\pi^2c^2m(v_-)^2$ is a force constant for nuclear distortion along this coordinate[24]. Although double exchange coupling has been estimated to be stronger than Heisenberg coupling in most [Fe$_2$S$_2$]$^+$ complexes[24,40], the combination of localizing contributions of Heisenberg exchange and vibrational coupling leads to partially delocalized $S = 1/2$ ground states (Robin-Day class II, Fig. 1d). For antiferromagnetically coupled mixed-valent dimers that display complete electron delocalization ($S = 9/2$, Robin-Day class III)[41], it has been shown that the large double exchange interaction prevails over the localizing effects of Heisenberg exchange and vibrational couplings (Fig. 1e)[24,42,43]. In our case, $^{57}$Fe Mössbauer experiments revealed two equally intense quadrupole doublets and isomer shift values for complexes **1** and **2**, consistent with a partially delocalized electronic structure (class II)[23]. Complex **3**, on the other hand, had only a single doublet, revealing a completely delocalized (class III) mixed-valent (Fe$^{2.5+}$-Fe$^{2.5+}$) electronic structure[23]. Previously, we proposed that the exchange of bridging chalcogens to heavier chalcogenide ions shifts the 'out-of-phase' PKS core vibration to lower energy. Therefore, we believe that a decreasing contribution of the PKS vibration plays a significant role in the ability to stabilize intermediate spin states for complexes **2** and **3**[23].

While previous EPR experiments revealed intermediate spin states and Mössbauer experiments established the extent of electron delocalization, vibronic coupling, and its influence were not directly studied. To experimentally probe these low-energy vibrations, we employed nuclear resonance vibrational spectroscopy (NRVS). NRVS is a synchrotron-based vibrational spectroscopy technique enabling selective observation of vibrational modes originating only from

Mössbauer active nuclei[44–46]. One of the most commonly employed nuclei for the NRVS experiments is $^{57}$Fe, which has proven to be particularly useful for analysis of complex, iron-containing biological systems, as it enables study of low-energy Fe normal modes without interference from complex environment contributions[47–51]. Besides $^{57}$Fe, other Mössbauer active nuclei can also be probed by the NRVS experiment. With brighter synchrotron sources capable of reaching the very high X-ray energies (>35 keV) needed for some Mössbauer transitions, and the development of high-resolution monochromators for these high energies[52–54], the ability and success of more exotic NRVS is now realized[16,55–60]. Of particular interest here is $^{125}$Te NRVS, as it has recently been successfully used for the identification of low-energy vibrational modes in Fe-Te clusters[16].

Replacement of the bridging S atoms with heavier Se atoms has previously been used to selectively probe local electronic structure in Fe-S proteins[13,14]. In a similar fashion, replacing bridging S atoms with heavier Te atoms unlocks the ability to perform both $^{57}$Fe and $^{125}$Te NRVS, which can offer us insight into the vibrations of Fe-Te clusters from two points of view, demonstrating the degree of iron or tellurium character in each mode. This site-selective advantage of NRVS spectroscopy is particularly beneficial to extract the Fe and Te core vibrations from the numerous vibrational modes involving mostly ligand, or the even more numerous protein background vibrations of Fe-Q metalloproteins. Previously, it was demonstrated that both $^{57}$Fe and $^{125}$Te NRVS can be used to understand the local structure of a $^{57}$Fe and $^{125}$Te in a labeled [4Fe-4Te]$^+$ cluster[16]. The $^{125}$Te labeling offered a different perspective, enabling the extraction of weak breathing modes in the $^{125}$Te NRVS that were otherwise unobservable in the $^{57}$Fe NRVS.

Herein, we employed $^{57}$Fe NRVS to study the influence of chalcogen identity on vibrational dynamics of the series of mixed-valent diiron dichalcogenide clusters **1**–**3**. The tellurium-bridged complex, **3**, was also studied by $^{125}$Te NRVS to provide insight into predominantly tellurium-based vibrational modes. $^{125}$Te NRVS of the oxidized form of this complex was also collected to explore the applicability of this technique for studying electron transfer processes in these types of clusters. NRVS measurements were complemented by DFT calculations to provide insight into the nature of observed vibrational modes and help with their assignment.

## Results and discussion
### $^{57}$Fe and $^{125}$Te NRVS results
$^{57}$Fe NRVS was used to characterize complexes **1**–**3**. Figure 2 shows the partial vibrational density of states (PVDOS) obtained in $^{57}$Fe NRVS experiments. The spectrum of complex **1** shows a range of transitions up to 400 cm$^{-1}$ with the most dominant transition centered at 139 cm$^{-1}$, overlapping with moderately intense transitions at 112 and 164 cm$^{-1}$. At higher energies, a doublet at 214 cm$^{-1}$ and a range of transitions between 280–400 cm$^{-1}$ can be observed. Higher energy transitions (380, 399 cm$^{-1}$) are expected to have significant Fe-S stretching character, due to the similarities with previously characterized Fe-S clusters that display transitions in the same region[61–67]. Considering a simple harmonic oscillator model for the Fe-Q vibrations (with the same force constant for all complexes), the transitions corresponding to the Fe-Q stretching vibrations are expected to shift toward lower energies with increased chalcogen mass (by a factor of 0.79 for Se and 0.72 for Te, compared to S)[16]. This indicates that the higher energy transitions in the spectrum of **1** (doublet between 380 and 400 cm$^{-1}$) should shift to 300–315 cm$^{-1}$ for **2** and 275–290 cm$^{-1}$ for **3**. Indeed, the $^{57}$Fe NRVS spectra of complexes **2** and **3** exhibit this expected trend as the highest energy transitions are centered ~303 cm$^{-1}$ for **2** and 246/294 cm$^{-1}$ for **3**. The influence of Q's mass is further observed in the differences between the $^{57}$Fe NRVS of **2** and **3** in the range of 220–310 cm$^{-1}$. For **2**, only a single broad peak is observed at 303 cm$^{-1}$, whereas two distinct peaks at 246 and 294 cm$^{-1}$ are observed for **3**. Given the change in mass and the relative intensity differences observed for the ~300 cm$^{-1}$

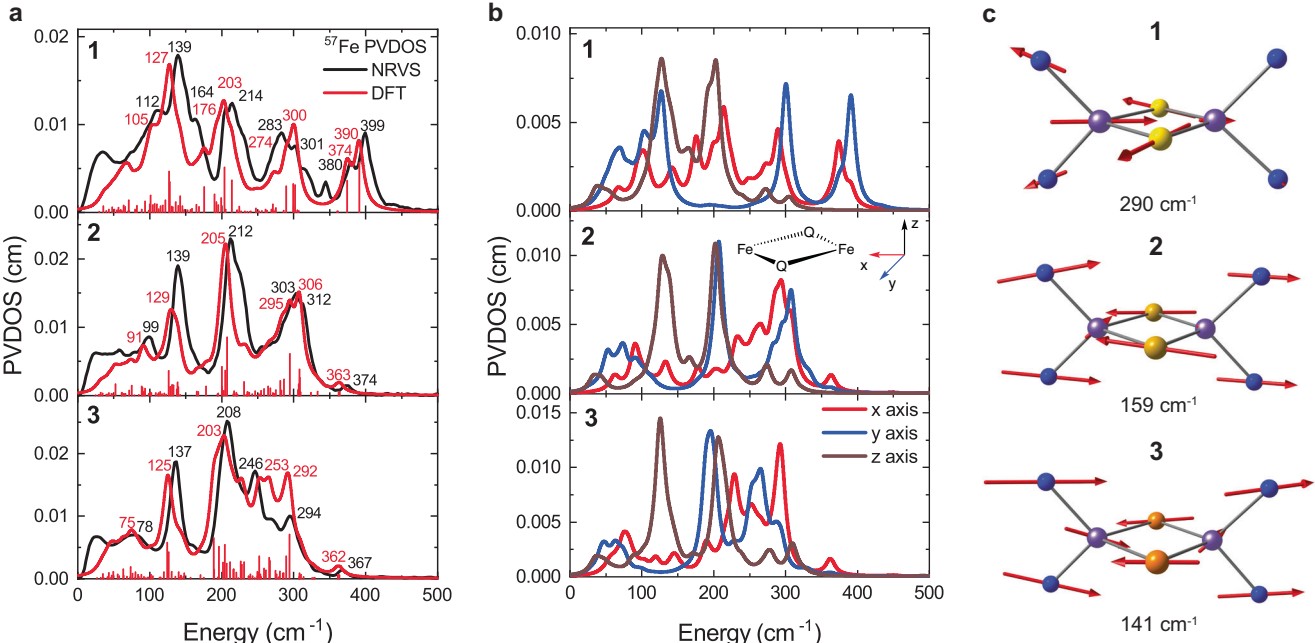

**Fig. 3 | Calculated $^{57}$Fe NRVS and PKS vibrational modes. a** Comparison of experimental (black line) and calculated (red line) $^{57}$Fe NRVS spectra of complexes **1–3**. Individual transitions are depicted by red vertical lines. **b** $^{57}$Fe PVDOS along $x$ (red line), $y$ (blue line), and $z$ (brown line) axes, for complexes **1–3**. The $x$ axis is defined along Fe-Fe bond, and $z$ axis is perpendicular to the [Fe$_2$Q$_2$]$^+$ core. **c** Representation of calculated PKS normal modes for complexes **1–3**. For clarity, only the [Fe$_2$Q$_2$]$^+$ cores and coordinating N atoms are shown.

features in the $^{57}$Fe NRVS spectra of **2** and **3** (Fig. 2A), we assign the 246 cm$^{-1}$ feature in **3** as having significant Fe-Te character. This feature is shifted significantly from the 294 cm$^{-1}$ feature in the spectrum of **3**. Conversely, the same Fe-Se feature for **2** is therefore overlapping significantly in the 303 cm$^{-1}$ feature and is unresolved from the other Fe vibrations. The transition at 112 cm$^{-1}$ in **1** also shifts significantly toward lower energies for **2** (99 cm$^{-1}$) and **3** (78 cm$^{-1}$) with lowering intensity of band, indicating that this vibrational mode also has significant chalcogen contribution (Fig. 2B). Unlike these features, transitions around 140 and 210 cm$^{-1}$ are most intense features in all spectra and remain mostly unchanged with changing mass of chalcogen, which suggests that these transitions correspond to the normal modes with significant Fe and small chalcogen displacements.

Complex **3** was also examined by a $^{125}$Te NRVS measurement. Figure 2 compares the $^{57}$Fe and $^{125}$Te PVDOS for complex **3**, and reveals a significantly different spectrum with the most intense $^{125}$Te features localized below 100 cm$^{-1}$, corresponding to the largest displacements of the bridging Te atoms. The most intense features in the $^{57}$Fe spectrum (200–300 cm$^{-1}$) have significantly lower relative intensity in the $^{125}$Te spectrum (compared to the most intense features), indicating smaller displacement of Te atoms in these modes, and suggesting that these modes are predominantly Fe-based, with minor Te character. The presence of a band around 250 cm$^{-1}$ in both $^{57}$Fe and $^{125}$Te spectra suggests that this band indeed has significant Fe-Te character, as suggested from the initial analysis of $^{57}$Fe spectrum. The larger mass of Te compared to S leads to smaller displacements of the Te atoms in Fe-Q stretching vibrations, which would also result in lower intensity transitions in the 220–300 cm$^{-1}$ region. Such observed transition localization is in agreement with a previously characterized 4Fe-4Te cluster[16].

$^{125}$Te NRVS was also used to characterize the oxidized [Fe$_2$Te$_2$]$^{2+}$ (**4**) complex. Differences between the $^{125}$Te NRVS of **3** and **4** are related to the increased Fe oxidation state and help further refine the nature of observed normal modes in $^{125}$Te NRVS spectra. Both **3** and **4** have similarities in the region below 200 cm$^{-1}$ (Fig. S4), indicating that oxidation state changes on the Fe centers do not have a significant influence on the predominantly Te-based vibrations (below 200 cm$^{-1}$). However, major differences are observed above 200 cm$^{-1}$, attributed to predominantly Fe-based vibrations, where the spectrum of **4** extends to approximately 310 cm$^{-1}$, reflecting the increased Fe oxidation. Shifting of the spectral features to the higher energies is in accordance with observed changes when altering Fe oxidation states in Fe-S complexes[68,69].

Experimentally, the collection of the $^{125}$Te NRVS is quite a bit more challenging. The determined 0 K Lamb-Mössbauer factor ($f_{LM}$) for the $^{125}$Te nuclear forward scattering was 0.49, significantly less than the $^{57}$Fe $f_{LM}$ of 0.87. While NRVS measures the energy loss from recoilless absorption and emission (1-$f_{LM}$), the signal-to-noise ratio of $^{125}$Te NRVS is notably lower than $^{57}$Fe, despite having a larger 1-$f_{LM}$ factor. This apparent paradox stems from the larger energy difference between the recoil energy and Debye energy of $^{125}$Te, which results in more multiphonon contributions compared to the predominantly single-phonon character of $^{57}$Fe experiments[57]. Although this multiphonon character significantly reduces data collection efficiency, distinct vibrations via $^{125}$Te NRVS remain resolvable, as demonstrated in our data and by others studying $^{125}$Te lattices[57].

## $^{57}$Fe NRVS DFT results

Density functional theory (DFT) calculations utilizing the broken symmetry (BS) approach ($S = 1/2$) were used to obtain information about the nature of the observed NRVS transitions. Despite the inability of BS DFT to optimize the geometry of an intermediate spin state, like the $S = 3/2$ ground state of complexes **2** and **3**, the $S = 1/2$ BS DFT optimized structures show good agreement with crystallographic data, indicating that the employed model can be further used (Figs. S5–S7, Table S1). Although there are ab initio methods that demonstrate the ability to calculate intermediate spin states[70,71] they often rely on truncated models and BS DFT geometry-optimized structures.

The calculated $^{57}$Fe NRVS spectra for all complexes show good agreement with experimental data, with major features deviating less than 15 cm$^{-1}$ from the experimental values (Figs. 3a and S8), further

validating the usage of the BS DFT approach for studying these complexes. For completeness, we also calculated the $S = 9/2$ solutions for **1**–**3** and the corresponding NRVS spectra to more faithfully represent a fully electronically delocalized structure **3** (Figs. S9–S10). The $S = 9/2$ solution has much worse agreement with the experiment for **1** and does not reproduce the splitting of the high-energy features. For **2** and **3**, however, the $^{57}Fe$ NRVS spectra of the $S = 1/2$ and $9/2$ solutions are more qualitatively like one another. As discussed later, though, the $S = 1/2$ solution has better agreement with the core's geometric structure, and the $^{125}Te$ NRVS also has better agreement with the experimental data (Fig. S10). This shows that the vibrational structure of **1** reflects a localized electronic structure; however, the distinction between localized vs. delocalized for **2** and **3** is much more challenging from the NRVS data alone, but the BS DFT calculated spectra have overall good agreement.

The Fe mode composition factors were analyzed along the three molecular axes (x axis is defined along the Fe-Fe bond and z axis is perpendicular to the $[Fe_2Q_2]^+$ plane). Figure 3b shows that for complex **1** high-energy transitions (in 250–400 $cm^{-1}$ range) are predominantly localized in the xy plane, and thus are dominated by in-plane Fe vibrations influenced by the nature of the bridging Q atom and the Fe-Q bond length. Some features at lower energies also show significant contributions of in-plane vibrations, but features around 130 and 200 $cm^{-1}$ are dominated by out-of-plane Fe vibrations (along z axis). High energy in-plane vibrations of complex **1** (250–400 $cm^{-1}$) shift to lower energies for complexes **2** (200–350 $cm^{-1}$) and **3** (160–330 $cm^{-1}$). Additionally, for complex **1**, the region below 250 $cm^{-1}$ shows significant contributions of in-plane Fe vibrations, while for complexes **2** and **3** these contributions are mostly localized below 100 $cm^{-1}$. Uniquely, the out-of-plane (z) vibrations remain mostly unchanged for all complexes (features around 130 and 200 $cm^{-1}$), demonstrating that changes in bridging chalcogen are responsible for the shifting of in-plane Fe vibrational modes, while minimally impacting the out-of-plane Fe vibrational modes.

To gain more insight into the nature of observed vibrations as a function of pairs of atomic displacements, the DFT-based kinetic energy distribution (KED) spectra were calculated for all complexes (Fig. S11). The KED spectrum for complex **1** shows that the high-energy transitions above 350 $cm^{-1}$ are the result of strong Fe-S vibrations with significant character along the S-S vector, and the region between 150 and 300 $cm^{-1}$ is dominated by vibrations with significant Fe-Fe character. The KED spectra of complexes **2** and **3** show significant shifts and separation for the Fe-Q and Q-Q vibrations, in comparison with **1**, potentially enabling a clearer observation of these vibrational modes in NRVS spectrum for **2** and **3**. Unlike **1**, the KED spectra of complexes **2** and **3** are dominated by Fe-Q vibration profile, and the similarity of this profile with the $^{57}Fe$ NRVS spectra of **2** and **3** indicates that these normal modes are Fe-dominated and that these vibrations involve small Se (for **2**) and Te (for **3**) nuclei displacements. This suggests that vibrational couplings between Fe and Q for complexes **2** and **3** are much weaker than for complex **1**. This is better illustrated by comparing calculated $^{57}Fe$ and Q ($^{32}S$, $^{80}Se$, $^{125}Te$) PVDOS spectra (Fig. S12), where $^{32}S$ and $^{80}Se$ are fictitious NRVS spectra as no Mössbauer active isotopes exist for these elements. Calculated PVDOS spectra of $^{57}Fe$ and $^{32}S$ for complex **1** have a similar profile, indicating strong vibrational coupling. Contrastingly, the spectra of $^{57}Fe$ and Q for complexes **2** ($^{80}Se$) and **3** ($^{125}Te$) have significantly different profiles, indicating weaker coupling between Fe and Q in these complexes.

### $^{57}Fe$ NRVS normal mode analysis

Individual key vibrations are best identified by analysis and visualization of individual normal modes and their relative intensity contributions to the NRVS spectrum. For complex **1**, the highest energy transitions (374 and 390 $cm^{-1}$) have a significant contribution of Fe-S bond stretching vibrations (Fig. S13). The features around 300 $cm^{-1}$ (of

complex **1**) are dominated by in-plane Fe-S core vibrations (Fig. S13), as well as out-of-phase in-plane breathing mode (PKS vibration, Fig. 3c), with additional Fe-S stretching vibrations toward lower energies, around 260 $cm^{-1}$. Transitions around 200 and 140 $cm^{-1}$ are dominated by strong Fe out-of-plane vibrations (Fig. S14), as suggested by the calculated polarized NRVS spectrum. The spectral feature at 112 $cm^{-1}$ in the experimental spectrum is also well reproduced with a shoulder at 102 $cm^{-1}$ in the calculated spectrum, and the normal mode that contributes most to this transition is the $[Fe_2S_2]^+$ core in-plane normal mode (Fig. S13).

The Fe-Q bond stretching vibrations shift from 370–390 $cm^{-1}$ in **1** to 280–310 $cm^{-1}$ in **2**, and further to 250–270 $cm^{-1}$ in **3**, consistent with increasing chalcogen mass (Fig. S13). Similarly, the in-plane $[Fe_2Q_2]^+$ core normal mode at 101 $cm^{-1}$ in **1**, shifts to 89 $cm^{-1}$ in **2** and 84 $cm^{-1}$ in **3** (Fig. S13). Other in-plane vibrations around 300 $cm^{-1}$ in **1** also shift to around 200 $cm^{-1}$ in **2** and **3** (Fig. S13) and overlap with the strong Fe out-of-plane vibrations around 200 $cm^{-1}$ (Fig. S14). The out-of-plane Fe vibrations around 140 and 200 $cm^{-1}$ remain constant among the series as these modes are not strongly influenced by changing the chalcogen mass (Fig. S14). The sharp feature at 246 $cm^{-1}$ in the experimental spectrum of complex **3** is well reproduced by a Fe-Te stretching band at 253 $cm^{-1}$ (Fig. S13), and the clearly resolved band at 294 $cm^{-1}$ is well reproduced by a Fe-Fe stretching vibration band at 292 $cm^{-1}$ (Fig. S13). Analogous Fe-Fe stretching modes in **1** (~220 $cm^{-1}$) and **2** (~260 $cm^{-1}$) are obscured by other features (Fig. S13). The reduction of the vibronic coupling between Fe and Q and resultant contraction of the spectra as the mass of Q increases leads to simpler, more resolved spectra with less contributions from Q. For complex **3**, this enables the observation of vibrations that are otherwise obscured in complexes **1** and **2**, highlighting the advantage of replacing S and Se with Te for characterization of normal modes in these complexes.

Identifying the out-of-phase breathing (PKS) mode, which leads to electronic localization in these complexes, is important for understanding the electronic structure of these mixed-valent $[Fe_2Q_2]^+$ clusters. The agreement between calculated and experimental spectra supports the DFT model's reliability in estimating the PKS vibration energy. DFT calculations predict a PKS vibration at 290 $cm^{-1}$ for **1** (Fig. 3c). The unambiguous assignment of the experimental peak at 301 $cm^{-1}$ to the PKS vibration is not possible due to the overlap of other equally intense in-plane Fe-S core vibrations (Fig. S15) calculated near the same energy and within the resolution of the experiment. Increasing chalcogen mass reduces the PKS mode energy to 158 $cm^{-1}$ in **2** and 141 $cm^{-1}$ in **3**, consistent with weaker vibronic coupling. Unfortunately, the resolution of the experimental $^{57}Fe$ NRVS spectra prevent direct observation of the PKS vibration, motivating further investigations via $^{125}Te$ NRVS for **3**.

### $^{125}Te$ NRVS DFT results

The calculated $^{125}Te$ NRVS spectrum shows overall good agreement with the experimental data (Fig. 4a, S16). Weak transitions in the $^{57}Fe$ NRVS spectrum, below 100 $cm^{-1}$, are the most intense transitions in the $^{125}Te$ NRVS spectrum, reflecting their significant Te character and providing an alternative view of the core vibrations compared to the $^{57}Fe$ NRVS experiment. In contrast to the strongest transitions in $^{57}Fe$ NRVS spectrum between 200 and 300 $cm^{-1}$, transitions in this region are weaker and less resolved in the $^{125}Te$ NRVS spectrum, indicating lower Te character in these transitions.

Analysis of the $^{125}Te$ mode composition factors along its molecular axes reveals that transitions below 50 $cm^{-1}$ are dominated by out-of-plane Te vibrations (Fig. S17). Transitions in the range 50–100 $cm^{-1}$ mostly consist of in-plane Te vibrations with significant out-of-plane character, while transitions above 100 $cm^{-1}$ are dominated by in-plane Te vibrations. The KED spectrum of complex **3** (Fig. S11) shows that the transitions at 118 and 156 $cm^{-1}$ have significant Te-Te character, and in general, unlike $^{57}Fe$ NRVS, $^{125}Te$ NRVS spectrum is better described by

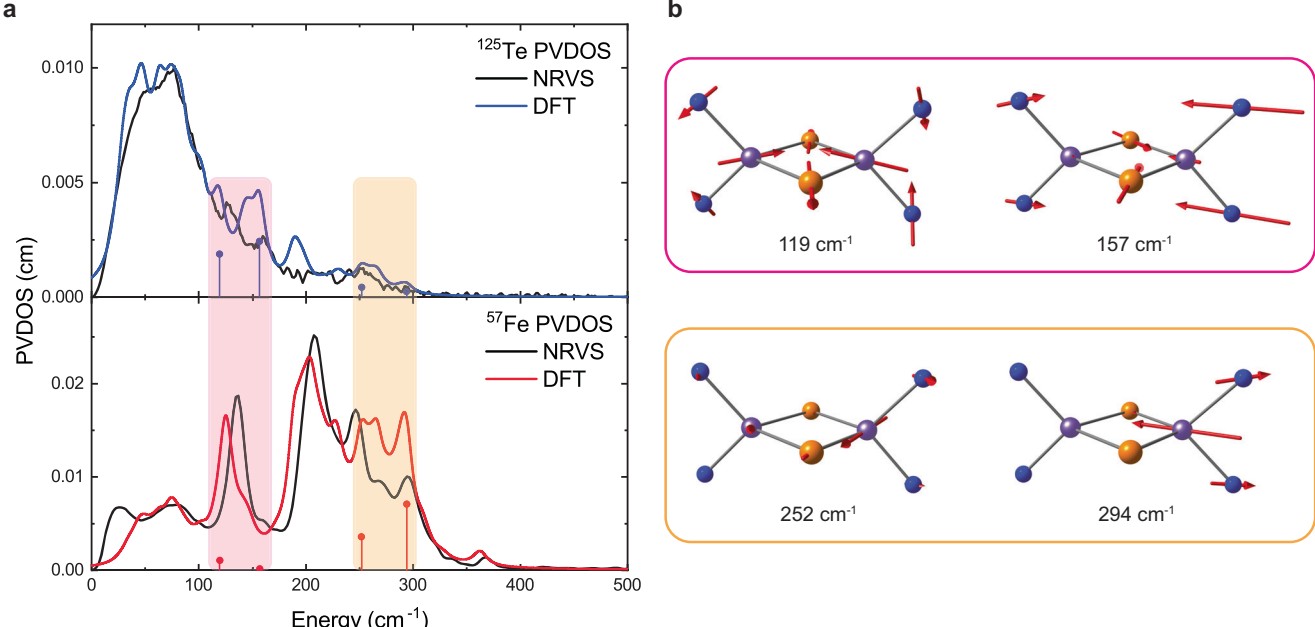

**Fig. 4 | Comparison of $^{57}$Fe and $^{125}$Te NRVS and selected vibrational modes.**
**a** Comparison of $^{125}$Te (top) and $^{57}$Fe (bottom) NRVS spectra of complex **3**.
Experimental spectra are depicted by black lines, calculated spectra by blue ($^{125}$Te)
and red ($^{57}$Fe) lines, with selected individual transitions depicted by blue and red
vertical lines. **b** Representations of selected calculated normal modes of complex **3**.

For clarity, only the [Fe$_2$Te$_2$]$^+$ core and coordinating N atoms are shown. Magenta
colored rectangles represent regions containing normal modes with significant Te
displacements (**a**) and corresponding normal modes (**b**). Orange colored rec-
tangles represent regions containing normal modes with significant Fe displace-
ments (**a**) and corresponding normal modes (**b**).

the Te-Te KED profile. The sharp band in experimental spectrum at
126 cm$^{-1}$ is well reproduced by the band at 118 cm$^{-1}$ (with the strongest
individual transition at 119 cm$^{-1}$) in the calculated spectrum (Fig. 4a).
Visualization of this mode shows that this band can be attributed to the
Te in-plane breathing mode (Fig. 4b). The experimental band at
160 cm$^{-1}$ is also well reproduced with the calculated band at 156 cm$^{-1}$
(with strongest individual transition at 157 cm$^{-1}$) and this transition can
be assigned to a Te-Te stretching vibration (Fig. 4b). Such an assign-
ment agrees with KED analysis which indicated that both bands have
significant Te-Te character.

At higher energies, the experimental spectrum shows weak broad
transition centered at 252 cm$^{-1}$ which was previously assigned from
$^{57}$Fe NRVS to a Fe-Te stretching vibration (Fig. 4b). It can be noted that
the same feature has greater relative intensity in $^{57}$Fe NRVS spectrum,
due to significant involvement of Fe nuclei in this normal mode
(Fig. 4a). Also, the intense Fe-Fe stretching mode at 294 cm$^{-1}$ in $^{57}$Fe
NRVS spectrum of **3** (Fig. 4a) can also be observed in the DFT
calculated $^{125}$Te NRVS spectrum as a very weak transition. The PKS
vibration of **3** overlaps with other strong, close-in-energy vibrations, in
this case, the strong Te-Te stretching vibration at 156 cm$^{-1}$, as seen in
the slightly overlapping peaks of the x- (PKS) and y- (Te-Te stretch)
polarized spectra (Fig. S17).

DFT calculations also reproduce the $^{125}$Te NRVS spectrum of **4** well
(Fig. S18), and polarized analysis of the $^{125}$Te composition factors show
the same trends as for complex **3** (Fig. S19). Overall, the same assign-
ments exist for the $^{125}$Te NRVS of **4**, as described for **3** above (see SI for
more detail). While predominantly Te-based vibrations are largely
unaffected by the Fe oxidation state change, vibrations with higher Fe
contributions show small shifts (~10 cm$^{-1}$) toward higher energies with
increasing Fe oxidation state, which is in agreement with observed
trends in experimental spectrum, as well as with expected changes
upon Fe oxidation. This further supports that $^{125}$Te can provide infor-
mation about electronic changes on neighboring Fe centers.

The direct comparison of $^{57}$Fe and $^{125}$Te NRVS spectra for complex
**3**, as well as intensities of selected normal modes in both spectra

(Fig. 4a), highlights the advantages of each of these techniques and
emphasizes the dual point-of-view of using both. The $^{57}$Fe NRVS best
reveals the direct observation of Fe-Te and Fe-Fe stretching vibrational
modes, while the $^{125}$Te NRVS spectrum enables identification of pre-
dominantly Te vibrations, whose intensities in the $^{57}$Fe spectrum are
too low to be observed and/or overlap with various other intense out-
of-plane Fe vibrations. Thus, the more selective characterization of the
Te vibrations via $^{125}$Te NRVS provides a clear advantage for character-
ization of Te containing complexes, enabling direct observation of
strong Te-Te vibrations that could not be identified in the $^{57}$Fe NRVS
experiments.

## Influence of vibronic coupling on the electronic structure
The vibronic couplings were obtained from the DFT calculated fre-
quencies of PKS normal modes, described above. For complexes **1–3**
an approximation for nuclear displacements in PKS normal mode
($\Delta r$) has been determined from comparison of their crystal struc-
tures with crystal structures of analogous oxidized complexes,
where a smaller displacement is observed for the heavier bridges
($\Delta r(\mathbf{1}) = 0.100$, $\Delta r(\mathbf{2}) = 0.067$, $\Delta r(\mathbf{3}) = 0.057$ Å)[23,72]. Complex **1** exhibits
a significantly larger vibronic coupling (2300 cm$^{-1}$) than complexes
**2** (610 cm$^{-1}$) and **3** (540 cm$^{-1}$), consistent with both the shifts of the
PKS vibration and decreased displacements of the heavier atoms.
Calculated vibronic couplings and previously experimentally esti-
mated values for $J$ (= −55 (**1**), −50 (**2**), −200 (**3**) cm$^{-1}$) and $B$ ( = 110 (**1**),
165 (**2**), 750 (**3**) cm$^{-1}$)[23] can be used to construct ground and excited
spin state potential energy surfaces in the PKS coordinate using
Eq. 2 (Fig. 5).

Calculated energy levels show that complex **1** displays a double-
well $S = 1/2$ ground state in agreement with the previous partially
delocalized class II assignment of the Robin-Day system (Fig. 5a)[23].
Complex **2** can also be characterized as partially delocalized (class II,
Fig. 5) with $S = 1/2$ and 3/2 states very close in energy, in agreement with
experimentally observed mixture of these states[23]. Unlike these,
complex **3** shows fully delocalized $S = 3/2$ ground state (class III, Fig. 5),

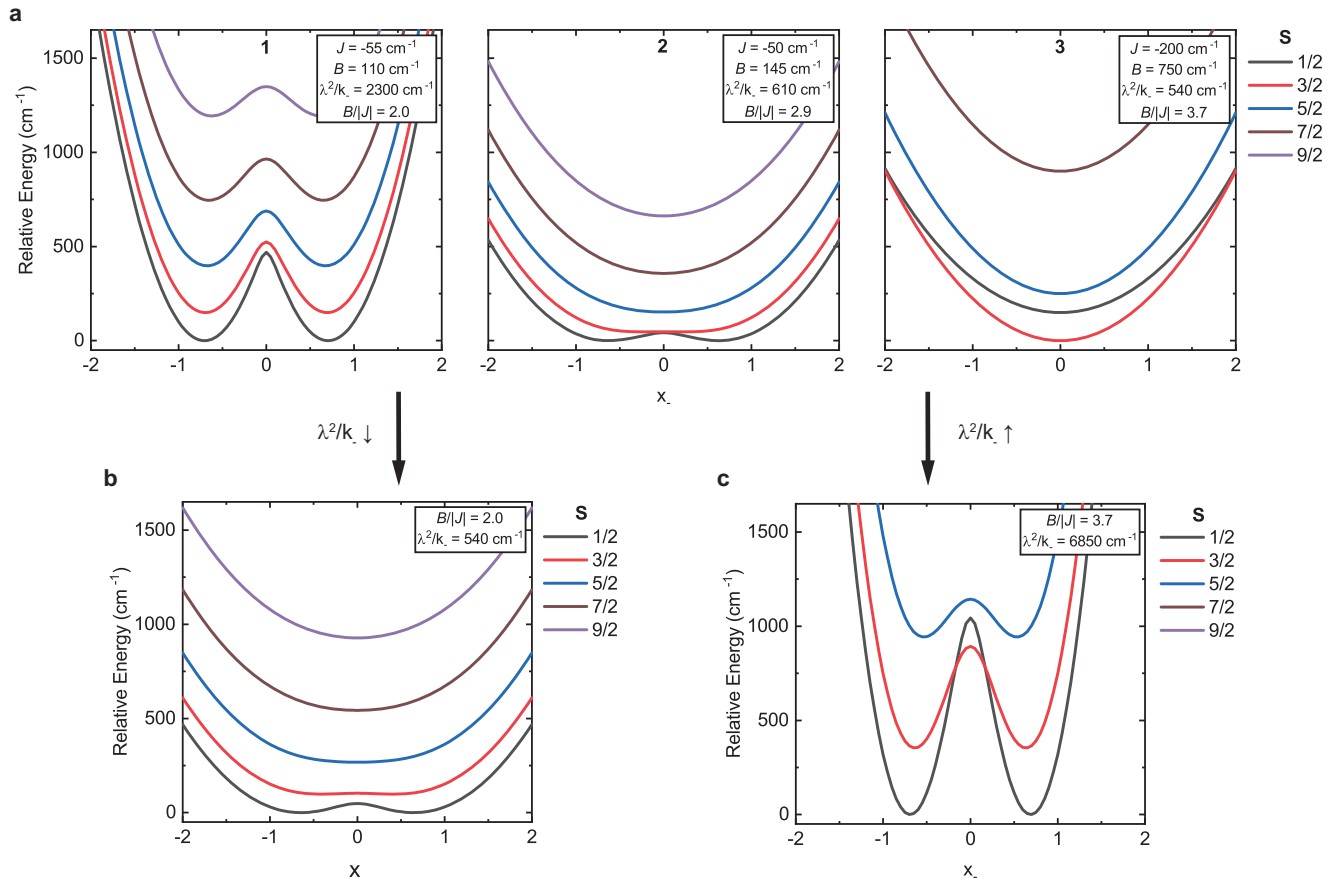

**Fig. 5 | Potential energy surfaces and influence of vibronic coupling. a** Ground and excited spin states in the PKS coordinate for the complexes **1**–**3**. **b** Spin states in the PKS coordinate for the complex **1** with a decreased amount of vibronic coupling. **c** Spin states in the PKS coordinate for the complex **3** with an increased amount of vibronic coupling.

which is also in line with experimentally observed intermediate $S = 3/2$ ground state[23].

To examine the influence of vibronic coupling on the electronic structure of these complexes, cases with varying vibronic coupling contribution (with other parameters constant) are considered (Figs. S22–S24). For complex **1**, lowering of vibronic coupling to the level observed for complex **3** results in lowering energies of excited states, but ground state remains partially delocalized $S = 1/2$ (Fig. 5a). Further lowering of vibronic coupling would lead to the completely delocalized ground state (class III, Fig. S22), which would still remain $S = 1/2$. While this indicates that vibronic coupling has a direct impact on the degree of electron delocalization, ground spin state should remain same for complex **1**. For complex **2**, increased vibronic coupling leads to a well-isolated ground spin state of $S = 1/2$, while decreased vibronic coupling may result in a delocalized $S = 1/2$ ground spin state with almost iso-energetic $S = 3/2$ excited state (Fig. S23). On the other hand, increasing vibronic coupling for complex **3** to the levels corresponding to the complex **1** result in a partially delocalized mixture of the $S = 1/2$ and $3/2$ states (similar to the scenario for complex **2**, Fig. S24), while a further increase results in larger separation of spin states and well isolated partially delocalized $S = 1/2$ ground state (class II, Fig. 5c). This analysis further confirms that vibrational coupling can have significant influence not only on the electron localization, but consideration of its contribution is crucial for the stabilization of intermediate spin states. For the studied complexes, it is shown that decreased vibronic coupling in complexes **2** and **3** leads to the stabilization of intermediate spin states. The analysis of Eq. 2 furthermore supports that the larger mass of Se and Te, and their decreased vibronic displacements in comparison to S is a major factor in the successful stabilization of these rare intermediate

spin states. This demonstrates that minimization of vibronic coupling results in the stabilization of excited states, and more accessible excited states should improve reactivity of the Fe-S clusters[70].

Understanding the vibronic coupling in mixed-valent transition metal cores is essential to our ability to design complexes and materials with tailored properties, including specified electronic spin states. In summary, we demonstrated here the successful application of [57]Fe NRVS to probe the vibrational dynamics of Fe centers in a series of diiron dichalcogenide complexes and identify normal modes with the help of DFT calculations. Additionally, it was shown that [125]Te NRVS can be employed as a complementary method to the [57]Fe NRVS, providing further information about low-energy Te-based normal modes. Such an approach enables studying these complexes from two different vantage points. Substitution with heavier chalcogens leads to contraction of [57]Fe NRVS spectra due to the smaller coupling between Fe center and bridging chalcogen. This results in spectra that have less influence from the chalcogen, leading to isolation of normal modes that could not be observed in complexes with lighter chalcogens. While the PKS vibration, responsible for electron localization in mixed-valent complexes, could not be directly identified in the experimental spectra, pairing with DFT calculations helps identify their signatures in the NRVS spectra and shows that the energy of this vibration decreases with heavier chalcogenides, consistent with smaller vibronic coupling between Fe and heavier chalcogens. These findings support that a decrease of the vibronic coupling in complexes with heavier chalcogens leads to the observed stabilization of intermediate spin states.

Although now two decades old, the application of NRVS spectroscopy to (bio)inorganic molecules is still developing, and extending the accessible nuclei beyond [57]Fe to [125]Te offers unique opportunities

for the characterization of iron chalcogenide clusters. While Te, unlike S and Se, is not widely present in biological systems, Te labeling of proteins has been successfully accomplished in the past and opens up the possibility of using Te as a probe to study biological systems[20,21,73–76]. In our attempts to understand vibronic contributions to the electronic structure of the mixed-valent diiron dichalcogenide cluster, $^{125}$Te offered the best extraction of important low-energy core vibrations. The systematic decrease in the contribution of the PKS to the electronic localization in the $[Fe_2Q_2]^+$ cluster, with increased chalcogenide mass, reveals that Se and Te substitutions in Fe-S clusters may not be strictly innocent. In this system, where the Fe-Fe distance does not change significantly, the dampening of the vibronics has allowed for double-exchanged intermediate spin states.

In other biological systems, such as nitrogenase which catalyzes reduction of $N_2$ to ammonia and contains FeMo cofactor ([7Fe9SMoC]) which consists of $[Fe_4S_3]$ and $[MoFe_3S_3]$ subunits bridged by three $\mu_2$ sulfides (belt sulfides) and one $\mu_6$ carbide[9,10], Se substitution of the belt sulfides under turnover conditions has been exploited as an atom and site-specific spectroscopic marker to understand the local electronic structure[12], particularly the redox level, of the neighboring iron ions, via X-ray absorption spectroscopy (XAS)[13]. While Se substitution appears innocent enough via XAS, more recent EPR spectroscopy of these Se-turnover samples exhibits redistributions of the rhombicity of the $S = 3/2$ zero-field splitting tensor, indicating a clear but perhaps not fully understood change in the electronic structure[15]. Our diiron dichalcogenide dimers further support that Se and further Te substitution can have significant electronic influences that are sometimes the result of a change in the delicate balances of Heisenberg, double exchange, and vibronic couplings.

## Methods

### General considerations

Unless indicated otherwise, all manipulations were performed using oven-dried glassware in an M-Braun nitrogen-atmosphere glovebox or on a Schlenk line using standard Schlenk techniques. Molecular sieves were activated by heating at 200 °C for 48 h under high vacuum. THF, toluene, diethyl ether, hexane, and pentane were purchased anhydrous from Sigma, further dried over sodium/benzophenone ketyl, vacuum-transferred before use, and stored over 4 Å molecular sieves. $KC_8$, $FeBr_2$, $C_{10}H_8$, $PCy_3$ (Cy = cyclohexyl), $PMe_3$ (1.0 M in PhMe), 2,6-diisopropylaniline, acetylacetone, $(Me_3Si)_2S$, $MeSO_3H$, $S_8$, $Se^0$, $Te^0$, naphthalene, concentrated HCl (37%), and n-BuLi (2.5 M in hexanes) were purchased from Sigma and used as received. $^{57}$Fe metal (≥96% $^{57}$Fe) was purchased from Campro Scientific GmbH (Berlin), and $^{125}$Te metal (88% $^{125}$Te) was purchased from US Services (Summit, NJ, USA) and used as received. $^1$H, $^{31}$P, and $^{125}$Te NMR spectra were recorded on a Bruker Avance III HD 500 NMR spectrometer.

### Synthesis of SePMe₃

The synthesis was adapted from the literature[77]. A solution of $Me_3P$ in PhMe (1.0 M, 14.0 ml, 14.0 mmol, 1.1 eq) was added to solid Se (0.9915 g, 12.6 mmol, 1.0 eq) in a 20 ml vial with rapid stirring. A moderate exothermic reaction took place, resulting in complete consumption of the Se solids. As the reaction cooled back to room temperature, white microcrystalline solids precipitated from the solution. After stirring for 12 hours at room temperature, the solids were collected on a glass frit and washed with pentane (12 ml) and dried under vacuum to afford 1.1921 g (ca. 61%) of the desired product. $^1$H and $^{31}$P data are consistent with reported values[77]. $^1$H NMR (CDCl₃), $\delta$ (ppm): 1.92 (d, 13 Hz, 9H, P(CH₃)₃). $^{31}$P NMR (CDCl₃), $\delta$ (ppm): 8.76 (s, SePMe₃, ca. 93% abundance (non-$^{77}$Se)), 8.78 (d, 677 Hz, $^{77}$SePMe₃, ca. 7% abundance $^{77}$Se).

### Synthesis of TePCy₃

The synthesis was adapted from the literature[78]. Solid $Te^0$ (0.3201 g, 2.51 mmol, 1.4 eq) was added to a solution of $PCy_3$ (0.5084 g,

1.81 mmol, 1.0 eq) in PhMe (10 ml) and the reaction was left to stir at room temperature for four days. The resulting suspension was filtered through a pad of celite to remove unreacted $Te^0$, and the resulting yellow filtrate was concentrated under vacuum. The residue was triturated with pentane (5 ml), collected on a glass frit, and washed with $Et_2O$ (10 ml). The resultant pale-yellow solid was dried under vacuum to afford 0.5262 g (ca. 71%) of the desired product. $^1$H and $^{31}$P data are consistent with reported values[78]. $^1$H NMR ($C_6D_6$), $\delta$ (ppm): 1.99–1.91 (m, 6H), 1.80–1.70 (m, 3H), 1.65–1.57 (m, 6H), 1.49–1.43 (m, 3H), 1.43–1.32 (m, 6H), 1.10–0.94 (m, 9H). $^{31}$P NMR ($C_6D_6$), $\delta$ (ppm): 27.73 (s, TePCy₃, ca. 94% abundance (non-$^{125}$Te)), 27.75 (d, $^1J(^{125}Te)$ = 1710 Hz, $^{125}$TePCy₃, ca. 6% abundance $^{125}$Te).

### Synthesis of OAsPh₃

The synthesis was adapted from the literature[79]. Under air, $H_2O_2$ (50%, 0.55 ml, 9.7 mmol, 1.3 eq) was added dropwise to an ice-water bath cooled solution of $AsPh_3$ (2.3 g, 7.5 mmol, 1.0 eq) in acetone (20 ml). Upon complete addition, a significant white precipitate formed. The mixture was removed from the ice bath, and upon warming, the precipitate resolubilized. After stirring for 1 hour at room temperature, the volatiles were removed under vacuum to afford a white microcrystalline residue. The residue was triturated with PhMe (5 ml), collected on a glass frit, and washed with additional PhMe (2 × 5 ml). The white microcrystalline product was then dried under vacuum to a weight of 2.3 g (95%). $^1$H NMR data are consistent with reported values[79]. $^1$H NMR (CDCl₃), $\delta$ (ppm): 7.65–7.55 (m, 6H, o-ArH), 7.45–7.36 (m, 9H, m,p-ArH).

### Synthesis of SAsPh₃

The synthesis was adapted from the literature[80]. In the glovebox, $(Me_3Si)_2S$ (1.50 ml, 7.11 mmol, 1.1 eq) was added dropwise to a stirring suspension of OAsPh₃ (2.0143 g, 6.25 mmol, 1.0 eq) in MeCN (12 ml). With each drop, a blue-green flash was observed, which quickly faded to yellow over one or two seconds. Upon complete addition, the OAsPh₃ had completely dissolved, and the resulting yellow solution was warm to the touch. After about 30–60 seconds, a large quantity of off-white microcrystals had precipitated from the solution. The reaction was left to stir overnight at room temperature. The precipitates were then collected on a glass frit and washed with $Et_2O$ (20 ml). The resultant off-white microcrystalline solid was then dried under vacuum to afford 1.7423 g (82%) yield of the desired product. $^1$H NMR data is consistent with reported values[81]. $^1$H NMR (CDCl₃), $\delta$ (ppm): 7.76–7.73 (m, 6H, o-ArH), 7.56–7.51 (m, 3H, p-ArH), 7.51–7.46 (m, 6H, m-ArH).

### Synthesis of LH

The synthesis was adapted from the literature[82]. 2,6-diisopropylaniline (23.5 ml, 0.125 mmol, 2.0 eq), acetylacetone (6.4 ml, 0.0623 mmol, 1.0 eq), and methanesulfonic acid (8.1 ml, 0.125 mmol, 2.0 eq) were combined with toluene (200 ml) in a 500 ml round-bottom flask charged with a stir bar and fitted with a Dean-Stark condenser. The reaction was then heated to reflux (135 °C, oil bath) for 12 hours. After cooling to room temperature, the volatiles were removed under vacuum via rotary evaporation until a viscous oil remained. A saturated solution of $Na_2CO_3$ (250 ml) was added to the residue, followed by $CHCl_3$ (150 ml). The resulting mixture was stirred for 5 min and then transferred to a 1000 ml separatory funnel. After mixing well, the layers were separated, and the aqueous phase was washed with additional $CHCl_3$ (2 ×100 ml). The combined organics were dried over $Na_2SO_4$, filtered, and then concentrated to a viscous oil via rotary evaporation. MeOH (150 ml) was added to the oil, resulting in the formation of a white crystalline solid. The mixture was heated to 60 °C for 50 minutes, then filtered hot on a glass frit. The solid was washed with additional MeOH (200 ml total) and then dried under vacuum to afford 15.4 g (ca. 59%) of the desired product. $^1$H NMR data is consistent with reported values[82]. $^1$H NMR (CDCl₃), $\delta$ (ppm): 12.15 (s, broad, 1H, OH), 7.16–7.13

(m, 6H, *m,o*-Ar*H*), 4.89 (s, 1H, (CNAr)$_2$C*H*), 3.14 (sept, 6.9 Hz, 1H, C*H*(CH$_3$)$_2$), 1.73 (s, 6H, *H*$_3$C(CNAr)$_2$CH), 1.23 (d, 6.9 Hz, 6H, CH(C*H*$_3$)$_2$), 1.14 (d, 6.9 Hz, 6H, CH(C*H*$_3$)$_2$).

## Synthesis of $^{57}$FeCl$_2$

The synthesis was adapted from the literature[83]. $^{57}$Fe metal (0.5021 g, 8.81 mmol) was added to a 50 ml round-bottom flask charged with a stir bar and fitted with an N$_2$ inlet and reflux condenser. Under N$_2$, concentrated HCl (37%, degassed, 10 ml) was added. The resulting mixture was heated to 100 °C overnight until H$_2$ evolution had ceased and visual inspection confirmed all of the solid $^{57}$Fe had been consumed. The solution was then cooled to room temperature and the volatiles removed under high vacuum (0.001 mbar). To remove the residual coordinated water, the residue was heated to 200 °C under high vacuum (0.001 mbar) for 2 hours, affording the desired anhydrous $^{57}$FeCl$_2$ as a light beige solid (1.0426 g, ca. 93%).

## Synthesis of LFe(PhMe)

The synthesis was adapted from the literature[84]. *n*-Butyllithium (2.0 ml, 5.0 mmol) was added dropwise to a solution of LH (2.0155 g, 4.81 mmol) in THF (10 ml). Upon complete addition, the pale-yellow solution was stirred for 30 min. Then FeBr$_2$ (1.0422 g, 4.83 mmol) was added and the reaction was stirred for 1 hour. All volatiles were then removed under vacuum and the residue was reconstituted with PhMe (10 ml). KC$_8$ (0.8344 g, 6.18 mmol) was added, and the mixture was left stirring for 72 hours at room temperature. The volatiles were then removed under vacuum, and the residue was extracted with hexane (3 × 10 ml). The hexane extracts were filtered through a glass microfiber pad and the filtrates concentrated under vacuum. The residue was again extracted with hexane (10 ml), filtered, and the filtrate reduced to half-volume under vacuum. The concentrated solution was then placed in the freezer (−35 °C) and stored overnight to afford dark-red crystals. The supernatant was pipetted off, again concentrated to half-volume and a second crop of crystals were grown overnight. Combined yield 1.3176 g (ca. 48%) of dark-red crystals.

## Synthesis of $^{125}$TePCy$_3$

In a Schlenk tube charged with a stir bar and fitted with a Teflon screw-in pin, $^{125}$Te metal (0.1926 g, 1.54 mmol) and excess PCy$_3$ (1.4847 g, 5.29 mmol) were combined in PhMe (20 ml). The Schlenk tube was sealed, removed from the glovebox, and heated to 100 °C in an oil bath with stirring for 16 hours. At this time, visual inspection revealed all the Te metal had been consumed. After cooling to room temperature, the Schlenk tube was brought back into the glovebox, the reaction was filtered through a glass microfiber pad, and volatiles were removed under vacuum from the filtrate. The resulting residue was triturated with Et$_2$O (20 ml), and the off-white solid collected on a glass frit. The solid was further washed with Et$_2$O to ensure excess PCy$_3$ was removed (ca. 60 ml total). The resulting solid was then collected and dried under vacuum to afford 0.4922 g (ca. 78%) of a pale-yellow powder. $^{31}$P NMR analysis revealed the major product to be $^{125}$TePCy$_3$ (ca. 80% by $^{31}$P NMR), with a small amount of OPCy$_3$ (ca. 5% by $^{31}$P NMR, $\delta = 46.8$ ppm) and an unidentified phosphorus-containing impurity (ca. 15% by $^{31}$P NMR, $\delta = 33.6$ ppm). $^{125}$Te NMR revealed $^{125}$TePCy$_3$ as the only tellurium-containing product ($\delta = -938.0$ ppm). The material was used without further purification. $^{31}$P NMR (C$_6$D$_6$), $\delta$ (ppm): 27.47 (d, $^1J(^{125}Te) = 1720$ Hz). $^{125}$Te NMR (C$_6$D$_6$), $\delta$ (ppm): −937.98 (d, $^1J(^{31}P) = 1710$ Hz).

## Synthesis of L$_2$Fe$_2$$^{125}$Te$_2$

The synthesis was adapted from the natural abundance isotopologue previously reported[23]. Filtered solutions of LFe(PhMe) (0.5461 g, 0.965 mmol, 1.0 eq) and $^{125}$TePCy$_3$ (0.4550 g, 1.12 mmol, 1.2 eq) in PhMe (4 ml each) were combined in a 20 ml scintillation vial, which was then capped and inverted several times to thoroughly mix the solutions.

Upon standing overnight, the mixture had produced dark brown precipitates, which were collected on a glass frit, washed with additional PhMe until the filtrates ran clean, and then dried under vacuum to afford 0.4518 g (ca. 39%) of the desired product as a maroon-black powder.

## Synthesis of [K(THF)$_5$][L$_2$Fe$_2$$^{125}$Te$_2$]

The synthesis was adapted from the natural abundance isotopologue previously reported[23]. A solution of KC$_{10}$H$_8$ was generated by stirring KC$_8$ (0.0737 g, 0.545 mmol, 1.7 eq) with naphthalene (0.0507 g, 0.396 mmol, 1.2 eq) in THF (3 ml). After 30 min, the dark-green KC$_{10}$H$_8$ solution was added dropwise to a stirring suspension of L$_2$Fe$_2$$^{125}$Te$_2$ (0.3866 g, 0.323 mmol, 1.0 eq) in THF (5 ml). Upon complete addition, the L$_2$Fe$_2$$^{125}$Te$_2$ had solubilized to give a dark-red solution. The reaction mixture was stirred for 1 h, and then all volatiles were removed under vacuum. The resulting dark residue was triturated with Et$_2$O (5 ml) and filtered through a glass microfibre filter pad. The resulting black-red solid was washed with additional Et$_2$O until the filtrate ran clear (ca. 5 ml). The black-red solid was then resolubilized in THF (ca. 5 ml), filtered through the glass microfibre pad, further concentrated to half-volume under vacuum, and stored at −35 °C to afford 0.1564 g (ca. 30%) of the desired product as black-red crystals.

## Synthesis of L$^{57}$Fe(PhMe)

The synthesis was adapted from the natural abundance isotopologue reported above. *n*-Butyllithium (2.0 ml, 5.0 mmol) was added dropwise to a solution of LH (2.0224 g, 4.83 mmol) in THF (10 ml). Upon complete addition, the pale-yellow solution was stirred for 30 min. Then, $^{57}$FeCl$_2$ (0.6063 g, 4.74 mmol) was added, and the reaction was stirred overnight at room temperature. All volatiles were then removed under vacuum, and the residue was reconstituted with PhMe (10 ml). KC$_8$ (0.8583 g, 6.35 mmol) was added, and the mixture was left stirring for 72 hours at room temperature. The volatiles were then removed under vacuum, and the residue was extracted with hexane (3 ×10 ml). The hexane extracts were filtered through a glass microfiber pad, and the filtrates concentrated under vacuum. The residue was again extracted with hexane (10 ml), filtered, and the filtrate reduced to half-volume under vacuum. The concentrated solution was then placed in the freezer (−35 °C) and stored overnight to afford dark-red crystals. The supernatant was pipetted off, again concentrated to half-volume, and a second crop of crystals was grown overnight. Combined yield 1.5621 g (ca. 58%) of dark-red crystals.

## Synthesis of L$_2$$^{57}$Fe$_2$S$_2$

The synthesis was adapted from the literature[23]. Filtered solutions of L$^{57}$Fe(PhMe) (0.5082 g, 0.897 mmol, 1.0 eq) and SAsPh$_3$ (0.2999 g, 0.886 mmol, 1.0 eq) in PhMe (4 ml each) were combined in a 20 ml scintillation vial, which was then capped and inverted several times to thoroughly mix the solutions. Upon standing overnight, the mixture had produced dark-red-brown precipitates, which were collected on a glass frit, washed with additional PhMe until the filtrates ran clean, and then dried under vacuum to afford 0.3365 g (ca. 75%) of the desired product as a burgundy powder.

## Synthesis of [K(THF)$_6$][L$_2$$^{57}$Fe$_2$S$_2$]•2THF

The synthesis was adapted from the literature[23]. A solution of KC$_{10}$H$_8$ was generated by stirring KC$_8$ (0.0553 g, 0.409 mmol, 1.7 eq) with naphthalene (0.0395 g, 0.308 mmol, 1.2 eq) in THF (3 ml). After 30 min, the dark-green KC$_{10}$H$_8$ solution was added dropwise to a stirring suspension of L$_2$$^{57}$Fe$_2$S$_2$ (0.2502 g, 0.247 mmol, 1.0 eq) in THF (5 ml). Upon complete addition, the L$_2$$^{57}$Fe$_2$S$_2$ had solubilized to give a dark-red solution. The reaction mixture was stirred for 1 h, and then all volatiles were removed under vacuum. The resulting dark residue was triturated with Et$_2$O (5 ml) and filtered through a glass microfibre filter pad. The resulting dark-red solid was washed with additional Et$_2$O until the

filtrate ran clear (ca. 5 ml). The red solid was then resolubilized in THF (ca. 5 ml), filtered through the glass microfibre pad, further concentrated to half-volume under vacuum and stored at −35 °C to afford 0.1886 g (ca. 47%) of the desired product as dark-red crystals.

## Synthesis of $L_2{}^{57}Fe_2Se_2$

The synthesis was adapted from the literature[23]. Filtered solutions of $L^{57}Fe(PhMe)$ (0.5222 g, 0.921 mmol, 1.0 eq) and $SePMe_3$ (0.1439 g, 0.928 mmol, 1.0 eq) in PhMe (4 ml each) were combined in a 20 ml scintillation vial, which was then capped and inverted several times to thoroughly mix the solutions. Upon standing overnight the mixture had produced dark-green precipitates, which were collected on a glass frit, washed with additional PhMe until the filtrates ran clean, and then dried under vacuum to afford 0.4165 g (ca. 41%) of the desired product as a dark-green powder.

## Synthesis of $[K(THF)_6][L_2{}^{57}Fe_2Se_2] \bullet 2THF$

The synthesis was adapted from the literature[23]. A solution of $KC_{10}H_8$ was generated by stirring $KC_8$ (0.0618 g, 0.457 mmol, 1.7 eq) with naphthalene (0.0465 g, 0.363 mmol, 1.35 eq) in THF (3 ml). After 30 min, the dark-green $KC_{10}H_8$ solution was added dropwise to a stirring suspension of $L_2{}^{57}Fe_2Se_2$ (0.2971 g, 0.268 mmol, 1.0 eq) in THF (5 ml). Upon complete addition, the $L_2{}^{57}Fe_2Se_2$ had solubilized to give a dark brown solution. The reaction mixture was stirred for 1 h, and then all volatiles were removed under vacuum. The resulting dark residue was triturated with $Et_2O$ (5 ml) and filtered through a glass microfibre filter pad. The resulting black solid was washed with additional $Et_2O$ until the filtrate ran clear (ca. 5 ml). The black solid was then resolubilized in THF (ca. 5 ml), filtered through the glass microfibre pad, further concentrated to half-volume under vacuum, and stored at −35 °C to afford 0.1977 g (ca. 43%) of the desired product as black crystals.

## Synthesis of $L_2{}^{57}Fe_2Te_2$

The synthesis was adapted from the natural abundance isotopologue previously reported[23]. Filtered solutions of $L^{57}Fe(PhMe)$ (0.5245 g, 0.925 mmol, 1.0 eq) and $TePCy_3$ (0.3847 g, 0.943 mmol, 1.0 eq) in PhMe (4 ml each) were combined in a 20 ml scintillation vial, which was then capped and inverted several times to thoroughly mix the solutions. Upon standing overnight, the mixture had produced dark brown precipitates, which were collected on a glass frit, washed with additional PhMe until the filtrates ran clean, and then dried under vacuum to afford 0.3891 g (ca. 70%) of the desired product as a black-brown powder.

## Synthesis of $[K(THF)_5][L_2{}^{57}Fe_2Te_2]$

The synthesis was adapted from the natural abundance isotopologue previously reported[23]. A solution of $KC_{10}H_8$ was generated by stirring $KC_8$ (0.0477 g, 0.353 mmol, 1.5 eq) with naphthalene (0.0384 g, 0.299 mmol, 1.25 eq) in THF (3 ml). After 30 min, the dark-green $KC_{10}H_8$ solution was added dropwise to a stirring suspension of $L_2{}^{57}Fe_2Te_2$ (0.2882 g, 0.239 mmol, 1.0 eq) in THF (5 ml). Upon complete addition, the $L_2{}^{57}Fe_2Te_2$ had solubilized to give a dark-red solution. The reaction mixture was stirred for 1 h, and then all volatiles were removed under vacuum. The resulting dark residue was triturated with $Et_2O$ (5 ml) and filtered through a glass microfibre filter pad. The resulting black-red solid was washed with additional $Et_2O$ until the filtrate ran clear (ca. 5 ml). The black-red solid was then resolubilized in THF (ca. 5 ml), filtered through the glass microfibre pad, further concentrated to half-volume under vacuum and stored at −35 °C to afford 0.2021 g (ca. 53%) of the desired product as black-red crystals.

## NRVS sample preparation

Samples for $^{57}Fe$ and $^{125}Te$ NRVS studies were prepared by grinding between 100 and 200 mg of material in an M-Braun nitrogen-atmosphere glovebox and placed into custom sample cells sealed with Kapton tape.

Samples were then flash frozen in liquid nitrogen and stored in liquid nitrogen until measurements. The frozen powder samples were transported to the beamlines in a dry-shipper cooled to liquid nitrogen temperatures.

## $^{57}Fe$ NRVS

$^{57}Fe$ NRVS data were collected at the synchrotron radiation facility SPring-8 at BL19LXU, operating in the C-mode bunch pattern with a 145.5 ns interval between x-ray pulses, as previously described[65]. The samples were placed in a helium flow cold finger cryostat maintained nominally at 10 K.

The x-ray beam after the undulator was monochromated at a high heat load Si(111) monochromator to ~1 eV bandwidth, and then further by the Ge(331) x2 Si(975) high-resolution monochromator (HRM) to ~0.8 meV linewidth centered at the nuclear resonance energy (~14.4 keV)[85]. A 2×2 avalanche photodiode (APD) array detector was used to detect the delayed $^{57}Fe$ nuclear fluorescence and Fe K fluorescence (from internal conversion) following the nuclear resonance excitation.

## $^{125}Te$ NRVS

$^{125}Te$ NRVS measurements were conducted at the Dynamics beamline P01 at PETRA III (DESY, Hamburg)[86] with a ring current of 100 mA in the 40 bunch mode, with a 192 ns separation between bunches. The X-ray beam was monochromatized to the $^{125}Te$ nuclear transition energy (35.5 keV) using a sapphire backscattering monochromator[54] with an energy resolution of 0.9 meV. The sample was placed in a closed-cycle cryostat and cooled to 22 K, with the temperature estimated via detailed balance analysis using Bose-Einstein statistics. This low temperature was selected to minimize multiphonon contributions, which are significant for $^{125}Te$ at room temperature. The nuclear resonance signal, contributing to the NRVS spectra, was separated in time from electronic X-ray fluorescence and scattering, and measured using a stack of two Si APD detectors (10 mm×10 mm), positioned close to the sample and detecting mainly the 27.4 keV internally converted nuclear fluorescence following the nuclear resonance excitation.

## DFT calculations

All calculations including geometry optimizations and analytical frequencies calculations were performed using ORCA version 5.04[87–89]. Experimentally obtained crystal structures of all studied complexes were used as starting points for geometry optimizations. Structures of all complexes were optimized without counter ion. Calculations were performed utilizing bp86 functional[90,91], along with atom-pairwise dispersion correction with the Becke-Johnson damping scheme (D3BJ)[92,93]. For the inclusion of relativistic effects zeroth-order regular approximation (ZORA)[94,95] was used along with relativistically contracted def2 Ahlrichs basis set[96,97]. Used basis sets included triple ζ ZORA-def2-tzvp basis set for Fe, N, S, Se atoms, double ζ ZORA-def2-svp basis set for C, H atoms and old-ZORA-TZVP basis set for Te atoms. The resolution of identity approximation for Coulomb integrals (RI)[98] was used to speed up calculations. To account for solvation effects, the conductor-like polarizable continuum model (CPCM)[99] with THF solvent was used. Calculation of antiferromagnetic ground states of studied complexes was achieved starting from ferromagnetic solution and using spin flip feature to produce broken symmetry (BS) solution.

DFT calculated NRVS spectra were represented as PVDOS, which is defined as[100,101]:

$$D_j(\bar{\nu}) = \sum_{\alpha} e_{j\alpha}^2 L(\bar{\nu} - \bar{\nu}_{\alpha}) \tag{3}$$

where $e_{j\alpha}^2$ represents mode composition factor of atom $j$ in normal mode $\alpha$, while $L(\bar{\nu} - \bar{\nu}_{\alpha})$ represents a line shape function. Normal mode composition factors represent a fraction of kinetic energy associated

with displacement, $r_{j\alpha}$, of nuclei $j$ with mass of $m_j$, in normal mode $\alpha$[100,102,103]:

$$e_{j\alpha}^2 = \frac{m_j r_{j\alpha}^2}{\sum_j m_j r_{j\alpha}^2} \qquad (4)$$

Calculated mode composition factors were broadened by convolution with a Lorentzian function, with 15 cm$^{-1}$ full width at half maximum (fwhm), to account for the resolution of NRVS experiments. The DFT calculated NRVS spectra were obtained by summing all broadened mode composition factors across all normal modes.

Normal mode composition factors were additionally analyzed as their projections along axes, where axes were defined such that x axis was along Fe-Fe bond, [Fe$_2$Q$_2$]$^{+/2+}$ core was in xy plane and z axis was perpendicular to [Fe$_2$Q$_2$]$^{+/2+}$ core. In that way normal modes involving displacements of specific nuclei along x and y axes can be attributed to in-plane vibrational modes of those nuclei, while normal modes involving displacements of specific nuclei along z axis are attributed to out-of-plane vibrations. Spectra displaying normal mode composition factors along different axes are obtained by broadening of calculated normal modes along axes by convolution with a Lorentzian function, with 15 cm$^{-1}$ fwhm, and summing all broadened mode composition factors across all normal modes.

For KED analysis, in terms of relative displacements of nuclear pairs, displacements of both nuclei were projected on the bond between nuclei of interest (4Fe-Q, 4Fe-N, 1 Fe-Fe, and 1 Te-Te bond), and mode composition factors were calculated by summing contributions of both nuclei. To focus on the normal modes with significant displacements along the bond between nuclei of interest, only modes in which the angle between the bond and nuclear displacements is less than 30° are considered. For creating KED spectra, normal mode composition factors were broadened by convolution with a Lorentzian function, with 15 cm$^{-1}$ FWHM and summed over all normal modes.

Chemcraft software[104] was used for visualization of individual normal modes as well as for comparison of experimental and optimized structures of studied complexes.

## Data availability
The data that support the conclusions of this study are either presented in the paper or in its Supplementary Information. Source data are provided with this paper, and all data are available from the corresponding author upon request. Source data are provided with this paper.

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

## Acknowledgements

The authors thank the Max Planck Society (GEC) and the National Institutes of Health (GM65440 to SPC). We acknowledge DESY (Hamburg, Germany), a member of the Helmholtz Association HGF, for the provision of experimental facilities. $^{125}$Te NRVS synchrotron radiation experiments were carried out at DESY at beamline P01. Beamtime was allocated for proposal I-20220762. $^{57}$Fe NRVS synchrotron radiation experiments were performed at BL19LXU in SPring-8 with the approval of RIKEN (Proposal No. 20230009). The authors thank the Max Planck Computing and Data Facility for allocation of high-performance computing time (on Raven).

## Author contributions

A.R. contributed to data analysis and the writing of the original draft. J.T.H. contributed to project conceptualization, sample synthesis, data acquisition, and manuscript editing. H.W., D.P., I.S., N.N., and Y.Y. each contributed to data collection and manuscript editing. S.P.C. contributed to conceptualization, manuscript editing, acquisition of resources, and funding acquisition. G.E.C. supervised and contributed to conceptualization, data analysis, writing of the original draft, acquisition of resources, and funding acquisition.

## Funding

## Competing interests

The authors declare no competing interests.
