## [Transparent Peer Review file · Nature Communications]

^{125}Te and ^{57}Fe Nuclear Resonance Vibrational Spectroscopic Characterization of Intermediate Spin State Mixed-Valent Dimers

Corresponding Author: Professor George Cutsail III

Version 0:

Reviewer comments:

Reviewer #1

(Remarks to the Author)

The manuscript by Radović et al describes the vibrational properties of diiron dichalcogenide clusters that vary in spin state population. The work experimentally tests the previously hypothesized conclusion in Henthorn et al. Nat. Chem. 14, 328-333 (2022) that decreased vibronic coupling leads to an increase in the $S = 3/2$ spin state as sulfur in the cluster is replaced by atoms with increasing mass. The results, in the form of ^{57}Fe and ^{125}Te NRVS along with DFT, support the prior hypothesis and shed insight as to how certain vibrational modes lend to the stabilization of intermediate spin states.

Compared to the previous Henthorn et al. work, the novelty lies in the application of NRVS to uncover vibrational modes, however the main conclusions for how vibronic coupling can influence electronic structure are somewhat similar. To this reviewer, the broader significance of the study on the impact of intermediate spin states is lacking. The discussion provides some insight as to how this might be important for select biological systems such as nitrogenase, however more discussion should be given as to why intermediate spin states and the properties that lead to their stabilization, are important for electron transfer processes or function.

I found the presentation of results and figures to be of high-quality, particularly the application of ^{125}Te NRVS is notable. Given the specialty of the technique, and the stated challenging aspect of the measurement on page 14, the authors should note if measurements were performed on replicate samples. A potential drawback of the study is the stated inability to apply the DFT/BS approach to the $S = 3/2$ intermediate state for complexes 2 and 3. More explanation should be given why this is the case or the authors should pursue other methodology. A $S = 9/2$ calculated spectrum should also be done for the ^{125}Te NRVS, similar to the ^{57}Fe NRVS case in Figure S8.

Other comments:

A sentence on the broader significance of the study should be included in the abstract and captured in the title if possible. Figure 1B legend should acknowledge reference #11 since this figure is nearly identical, albeit the coloring, to Figure 1A of that paper.

Sample concentrations for complexes 1-3 should be given in Figure 2 or indicated in the methods for ^{57}Fe NRVS and ^{125}Te NRVS measurements.

Figure 2, the spectrum of complex 1 displays a feature at 164 cm^{-1} that is not discussed, and it is not clear if this feature shifts in spectra of Se and Te derivative complexes.

Experimental details for how the $[\text{Fe}_2\text{Te}_2]^{2+}$ complex (4) was oxidized should be given. As an important control, verification of the extent of oxidation should be completed by a complimentary technique, such as EPR or Mössbauer.

In the discussion, reference to the nitrogenase "belt sulfides" is a bit obscure especially for the non-expert. More context and description of the FeMo cofactor, along with the reaction the enzyme catalyzes, should be given.

Reviewer #2

(Remarks to the Author)

Cutsail and coworkers synthesized and characterized a series of mixed-valent diiron dichalcogenide complexes, $[\text{L}_2\text{Fe}_2\text{Q}_2]^-$

(Q = S, Se, Te), revealing that heavier chalcogenides (Se, Te) stabilize unprecedented $S = 3/2$ spin states due to reduced vibronic coupling. Using ^{57}Fe and ^{125}Te NRVS, they probed vibrational dynamics, demonstrating weaker Fe-Q coupling with increasing chalcogen mass. DFT calculations corroborated experimental findings, showing a shift of PKS normal modes to lower energies, leading to decreased vibronic coupling and stabilization of intermediate spin states. Their results highlight the influence of heavier chalcogens on electronic structure, with implications for designing spin-state-tuned materials. There are the following comments that need to be addressed:-

- In the introduction, I suggest adding a few lines explaining why Se and Te were included in the study. While the discussion on sulfur-containing complexes is well-developed, the rationale for incorporating heavier chalcogenides is not explicitly mentioned. Clarifying this aspect would strengthen the motivation behind their inclusion.
 - The study emphasizes the role of heavier chalcogens in modulating spin states via vibrational effects, but what about the role of the β -diketiminato ligand scaffold? How does the steric or electronic tuning of these ligands impact exchange interactions and the extent of delocalization? Are there comparisons with other ligand frameworks to confirm the generality of the observed trends?
 - The explanation of NRVS spectra trends is quite detailed, but the text could benefit from better structuring. The discussion jumps between experimental data and theoretical calculations without clear transitions. It would be useful to introduce a logical flow—first presenting experimental results comprehensively, then introducing the computational study, and finally comparing both.
 - Some statements are repetitive, particularly when discussing peak shifts due to increasing chalcogen mass. The explanation of spectral shifts and in-plane vs. out-of-plane vibrations could be streamlined (I think the manuscript needs to be shortened a bit; it will benefit the readers).
 - The manuscript relies heavily on NRVS data to assign oxidation states and electronic structures. However, NRVS alone cannot definitively distinguish between localized and delocalized states. I recommend incorporating complementary techniques such as X-ray absorption spectroscopy (XAS) or Mössbauer spectroscopy to strengthen the conclusions regarding electronic localization and delocalization
 - Given the high-spin nature of these complexes, is single-determinant DFT sufficient for an accurate description?
 - The manuscript suggests that complex 2 has a more localized electronic structure than complex 3, but the claim lacks clear evidence, as the NRVS frequency shifts could be due to ligand field effects rather than electronic delocalization, and there is no supporting computational charge/spin density analysis; therefore, including spin density plots to directly visualize electron delocalization across the Fe centres would strengthen the argument.
 - I suggest that the author perform DFT optimizations considering multiple spin states, exploring all possible combinations of overall multiplicity. For example, in complex 1, one Fe centre could be treated as an $S=2$ site, while the other could be considered an $S=3/2$ site, leading to overall multiplicities of either $S=1/2$ or $S=9/2$. Such an approach would allow the author to determine whether the molecule exhibits a ferromagnetically coupled ground state ($S=9/2$) or a lower-spin state ($S=1/2$). Additionally, I recommend providing the DFT-computed spin ladders, i.e., the relative energies of the $S=1/2, 3/2, 5/2, 7/2,$ and $9/2$ spin states for all complexes, along with visualizations of spin density distributions for each solution. These findings would offer valuable insights into the electronic structure and magnetic behaviour of the mixed-valence system.
 - If the double exchange parameter (B) and magnetic exchange parameter (J) have already been reported, as cited in Figure 1, the author should explicitly state these values and further analyze the B/J ratio in relation to experimentally and computationally investigated vibronic coupling. It would be beneficial to include a separate discussion section that systematically highlights and compares these findings. Additionally, the author should clearly classify each molecule into mixed-valence Class I, II, or III, providing a rationale for these classifications. Furthermore, the discussion should emphasize the novel aspects of the study, particularly in the context of vibronic coupling, given that this concept has been previously explored.
 - The manuscript claims that chalcogen substitution ($\text{O} \rightarrow \text{S} \rightarrow \text{Se}$) significantly affects Fe oxidation states and electronic delocalization, but the justification is weak, as the only evidence provided—vibrational frequency shifts—may stem from mass effects rather than electronic changes, performing Natural Population Analysis (NPA) or Löwdin charge analysis, along with comparing experimental redox potentials, would provide stronger evidence.
 - The authors assume that the highest-energy peaks correspond to Fe-Q vibrations but do not validate this with isotopic substitution, I don't know if it could be done or not.
 - DFT geometry optimizations were performed using the BP86 functional in ORCA. However, at the molecular level, the widely used hybrid B3LYP functional has been found to provide more accurate structural corrections. I suggest performing a few optimizations using the B3LYP functional and comparing the results with experimental structural parameters. This comparison would help clarify whether the GGA functional, such as BP86, is indeed superior to the hybrid functional in this case. Also, I would suggest that the author include vibrational frequencies along with corresponding oscillator strength to verify the global minimum of the optimised geometries.
 - In Figure 2., Indeed the author is able to explain and match the Fe-Q vibrational with the already reported $\{\text{Fe-X}\}$ ($X = \text{Chalcogen}$) clusters. I would suggest providing a picture of all the vibrations (e.g. 139 cm^{-1} , 137 cm^{-1} , 112 cm^{-1} , 99 cm^{-1} , 283 cm^{-1} , etc.) showing the movement of atoms as given in Figures S11, S12 and S20. Interestingly, at lower energy vibrations, I can see a shoulder peak in molecule 1 (Figure 2) around 78 cm^{-1} , which downshifts in 2 to 75 cm^{-1} , and it disappears in the case of 3. It would be interesting if the author explains about it.
- Overall, I believe the manuscript in its current form doesn't meet the standard to be published in the Nat. Commun.

Reviewer #3

(Remarks to the Author)

The manuscript presents a study of dinuclear mixed-valent iron complexes with S, Se, and Te bridges. All three have been previously reported by some of the authors, and the interesting property of the Se and Te clusters is that they exhibit intermediate total spin of $S = 3/2$ as opposed to the "normal" $S = 1/2$ low-spin situation (ref. 11). This arises due to the

competition between localizing and delocalizing effects (vibronic coupling / double exchange). The manuscript approaches the subject using a combination of ^{57}Fe and ^{125}Te nuclear resonance vibrational spectroscopy (NRVS), a technique that can isolate vibrational modes associated with specific Mossbauer-active nuclei. In light of this work (as well as of ref. 46) the S/Te substitution as basis for NRVS, becomes a very important avenue for studying Fe/S systems in great local detail. The analysis is supported by computational models, which help in confirming the validity of the interpretations. For the present series of compounds the conclusions are quite clear, with a specific vibration (out of plane - "PKS") shown to become progressively dampened with heavier chalcogenides, allowing double exchange to dictate the intermediate spin state. Beyond the specific compounds, the study has exceptional value as an eloquent and compelling demonstration of how powerful this combined approach can be, with the Fe/Te-NRVS revealing distinct vibrations or enhancing different common vibrations in the low-energy part of the spectrum. The account of the research and the corresponding analysis are nothing less than a reference guide for how this approach is to be used and how chemically meaningful information can be extracted from such experiments.

Version 1:

Reviewer comments:

Reviewer #1

(Remarks to the Author)

The authors have adequately addressed my comments, and I am happy to support the revised manuscript for publication.

Reviewer #2

(Remarks to the Author)

Cutsail and coworkers synthesized and characterized a series of mixed-valent diiron dichalcogenide complexes, $[\text{L}_2\text{Fe}_2\text{Q}_2]^-$ (Q = S, Se, Te), revealing that heavier chalcogenides (Se, Te) stabilize unprecedented $S = 3/2$ spin states due to reduced vibronic coupling. Using ^{57}Fe and ^{125}Te NRVS, they probed vibrational dynamics, demonstrating weaker Fe-Q coupling with increasing chalcogen mass. DFT calculations corroborated experimental findings, showing a shift of PKS normal modes to lower energies, leading to decreased vibronic coupling and stabilization of intermediate spin states. Their results highlight the influence of heavier chalcogens on electronic structure, with implications for designing spin-state-tuned materials. In the revised manuscript, the authors have attempted to address my previous comments; however, I find that some of the responses remain unconvincing or do not fully resolve the concerns I had raised—

- (For comment 1) the response provided does not fully address my original concern. I believe the rationale for including Se and Te over S should be more clearly justified in terms of their impact on the electronic structure, rather than being implied through steric effects or ease of spectroscopic characterization. A brief explanation in the introduction highlighting the relevance of heavier chalcogens in tuning electronic properties—such as their ability to modulate metal–ligand covalency or influence spin state energetics—would significantly strengthen the scientific motivation for their inclusion in this study.
- While I understand the authors' preference for combining results and discussion, I believe that a clearer, section-wise comparison between the DFT and experimental findings—particularly in a side-by-side manner—would significantly enhance readability. Presenting the computational data alongside the corresponding experimental results within the Results section (rather than deferring comparisons to the general discussion) would allow the reader to more easily appreciate the alignment or discrepancies between the two. This structure would also reinforce the coherence of the NRVS spectral interpretation and avoid the current back-and-forth flow between theory and experiment. I still suggest reconsidering this presentation style to improve the manuscript's clarity and logical progression.
- I appreciate the authors' clarification regarding the primary objective of the manuscript and the prior characterization of the oxidation and spin states. However, even though these oxidation states have been thoroughly established in previous work using Mössbauer spectroscopy, X-ray emission, and EPR, I still believe it would benefit the reader if the manuscript included a concise sentence explicitly linking those earlier findings to the current study. A brief reference—perhaps in the introduction or results section—would help provide clearer context and reinforce how this vibrational study builds upon the previously established electronic structure. Including this one conclusive line would strengthen the narrative continuity for readers who may not be familiar with the prior work.
- While I acknowledge the authors' point that the ground states have been experimentally established in prior work, I still believe that incorporating DFT calculations for both the $S = 1/2$ and $S = 9/2$ states can offer valuable complementary insight—especially in the context of spin delocalization and magnetic exchange interactions. Since DFT studies on similar mixed-valent systems have already been reported, these calculations are feasible and can help visualize the spin density distribution across the metal centers. This, in turn, would clarify how ferro- and antiferromagnetic coupling contribute to the overall spin state. Therefore, I recommend that the authors report the relative DFT energies of the $S = 1/2$ and $S = 9/2$ states, considering both ferro- and antiferromagnetic alignments of the metal centers. Prior studies have shown that in mixed-valent systems, electronic oxidation or reduction can trigger a switch from antiferromagnetic to ferromagnetic coupling, and such a comparison here would be particularly relevant. Even if the absolute spin ladder energies from DFT are approximate, the qualitative trends can still provide meaningful insights that enrich the discussion of spin delocalization and electronic structure. (Dalton Trans., 2013, 42, 16490-16493, 10.1021/ja101887f, Phys. Chem. Chem. Phys., 2022, 24, 20760)
- In one of my earlier comments, what I intended to suggest was the following: select one of the most significant vibrational modes and use it to generate perturbed geometries along that mode. Then, compute the electronic structure for these displaced geometries to examine how the double exchange interaction varies with vibrational distortion. This approach would offer valuable insight into how specific vibrational motions influence double exchange and magnetic coupling. Even performing this analysis for just one representative molecule would allow for a meaningful magneto-structural correlation and provide a deeper understanding of the coupling between electronic and vibrational degrees of freedom. This is the core

idea I was aiming to convey in my earlier suggestion.

The study addresses an important topic, and with the incorporation of the suggested minor revisions—particularly those clarifying the connection to prior work and enhancing the theoretical analysis—the authors could come back with a significantly strengthened manuscript suitable for reconsideration.

Reviewer #3

(Remarks to the Author)

The revised version offers important clarifications on certain points that might have been ambiguous or that assumed solid knowledge of prior literature (e.g. with respect to the electronic structure and oxidation states of the systems under study) or familiarity with specific technical/methodological aspects (e.g. with respect to the limits of DFT applicability for intermediate spin states). In my view the authors sufficiently address or rebut all concerns raised on the initial submission.

Understandably, requests that had to do with revisiting and replicating prior work cannot be addressed. I still view this work as an exemplary demonstration of combined Fe/Te NRVS and I expect that this study will become a reference point for a niche but powerful methodology.

[Note from editor: Reviewer #3 was also asked to look over the authors' responses to Reviewer #2's comments]

Reviewer #1 (Remarks to the Author):

Comment: The manuscript by Radović et al describes the vibrational properties of diiron dichalcogenide clusters that vary in spin state population. The work experimentally tests the previously hypothesized conclusion in Henthorn et al. Nat. Chem. 14, 328-333 (2022) that decreased vibronic coupling leads to an increase in the $S = 3/2$ spin state as sulfur in the cluster is replaced by atoms with increasing mass. The results, in the form of ^{57}Fe and ^{125}Te NRVS along with DFT, support the prior hypothesis and shed insight as to how certain vibrational modes lend to the stabilization of intermediate spin states.

Compared to the previous Henthorn et al. work, the novelty lies in the application of NRVS to uncover vibrational modes, however the main conclusions for how vibronic coupling can influence electronic structure are somewhat similar. To this reviewer, the broader significance of the study on the impact of intermediate spin states is lacking. The discussion provides some insight as to how this might be important for select biological systems such as nitrogenase, however more discussion should be given as to why intermediate spin states and the properties that lead to their stabilization, are important for electron transfer processes or function.

Response: We thank the reviewer for their comments and efforts to review our manuscript. We are pleased to read that the reviewer finds the technique and approach novel. We appreciate that the reviewer understands the unique electronic structure of these complexes, as detailed in the previous work, and our attempts now to understand how vibronic coupling play a central role. Given the novelty of the work, both the spectroscopic aspects and electronic structure, we have taken the reviewer's suggestion and added more discussion of the importance and relevance of intermediate spin-states. In our expanded discussion, we have now elaborated more on the influence of the mass of the bridging atom and demonstrated that the stabilization of the intermediate spin-states is a direct result of the increased mass and decreased vibronic contributions to electronic localization.

Comment: I found the presentation of results and figures to be of high-quality, particularly the application of ^{125}Te NRVS is notable. Given the speciality of the technique, and the stated challenging aspect of the measurement on page 14, the authors should note if measurements were performed on replicate samples.

Response: We are pleased that the reviewer found our results and figures to be of high quality. Due to the relatively high cost of the isotopic labelling, we were unable to prepare replicate samples for study.

Comment: A potential drawback of the study is the stated inability to apply the DFT/BS approach to the $S = 3/2$ intermediate state for complexes 2 and 3. More explanation should be given why this is the case or the authors should pursue other methodology.

Response: While intermediate spin states ($S = 3/2$) cannot be described by DFT, we observe good agreement between experimental and calculated geometries. While computationally expensive multi-reference methods *may* be capable of performing geometry optimization for other desired spin-states, our approach and comparisons of the low- and high-spin BS-DFT solutions have clear distinctions.

We do note that others have successfully calculated the electronic structure of model Fe_2Te_2 cores (<https://doi.org/10.1039/D2CP02975H>), but their procedures also rely on using either the crystallographic geometry or the BS-DFT optimized geometry before proceeding to multi-reference calculations.

As the vibrational modes are inherently dependent on the structure, and the low-spin BS geometry optimization exhibits excellent agreement with the x-ray crystallography of the complex, we still believe our DFT calculations accurately represent the vibrational data.

Additional text has been added to the manuscript on page 9 to further discuss this point.

“Although there are ab initio methods that demonstrate the ability to calculate intermediate spin-states they often rely on the truncated models and BS DFT geometry optimized structures.”

Comment: A $S = 9/2$ calculated spectrum should also be done for the ^{125}Te NRVS, similar to the ^{57}Fe NRVS case in Figure S8.

Response: We thank the reviewer for the suggestion. The comparison of calculated ^{125}Te NRVS spectra for $S = 1/2$ and $S = 9/2$ states of **3** has been added to the SI (Figure S10). The low-spin geometry solution has better agreement with the experimental data, further supporting our use of it, as discussed above.

Other comments:

Comment: A sentence on the broader significance of the study should be included in the abstract and captured in the title if possible.

Response: We have added some text about the broader significance of the study as highlighted in the abstract.

“The findings suggest that heavy character of larger chalcogen atoms results in decreased vibronic coupling of electronic localization modes; the observation of an intermediate spin-state is shown to be unattainable for lighter Fe₂Q₂ cores. This highlights the crucial role of vibronic coupling in modulating the electronic structure of mixed-valence systems and should enhance understanding of the electronic structure in more complex biological Fe-S clusters. Furthermore, gaining insights into vibronic coupling could aid in the design of mixed-valent systems with tailored properties.”

Additionally the title has been changed to:

“¹²⁵Te and ⁵⁷Fe Nuclear Resonance Vibrational Spectroscopic Characterization of Diiron Dichalcogenide Core Vibrations and their Contributions to Intermediate Spin-States”

Comment: Figure 1B legend should acknowledge reference #11 since this figure is nearly identical, albeit the coloring, to Figure 1A of that paper.

Response: Reference added.

Comment: Sample concentrations for complexes 1-3 should be given in Figure 2 or indicated in the methods for ⁵⁷Fe NRVS and ¹²⁵Te NRVS measurements.

Response: Measurements were performed on solid samples. Additional information about sample preparation were added to the SI.

Comment: Figure 2, the spectrum of complex 1 displays a feature at 164 cm⁻¹ that is not discussed, and it is not clear if this feature shifts in spectra of Se and Te derivative complexes.

Response: The absence of similar feature around 160 cm⁻¹ in the spectra of complexes 2 and 3 would suggest that feature at 164 cm⁻¹ in 1 shift toward lower energies. Proximity of intense feature at 140 cm⁻¹ possibly prevents observation of this band in spectra of 2 and 3.

Comment: Experimental details for how the [Fe₂Te₂]²⁺ complex (4) was oxidized should be given. As an important control, verification of the extent of oxidation should be completed by a complimentary technique, such as EPR or Mössbauer.

Response: We believe the presentation order in the manuscript has perhaps misled the reviewer about the synthetic preparation. The oxidized cluster 4 is the synthetic precursor to the mixed valent reduced cluster 3. We have followed the same synthetic procedure as we have previously reported, and given the general reproducibility of the results, new Mössbauer data was not collected. Additionally, the drastic solubility difference between the oxidized and reduced complexes (the crystalline oxidized cluster has virtually zero solubility in commonly used solvents) ensures that there is no appreciable mixture in the two samples (i.e. there is no reduced cluster present in the oxidized sample, and there is no appreciable oxidized cluster in the reduced sample).

Comment: In the discussion, reference to the nitrogenase “belt sulfides” is a bit obscure especially for the non-expert. More context and description of the FeMo cofactor, along with the reaction the enzyme catalyzes, should be given.

Response: We have added additional text to the final paragraph to further elaborate on the role of nitrogenase and its structure:

“In other biological systems, such as nitrogenase which catalyzes reduction of N₂ to ammonia and contains FeMo cofactor ([7Fe9SMoC]) which consists of [Fe₄S₃] and [MoFe₃S₃] subunits bridged by three μ₂ sulfides (belt sulfides) and one μ₆ carbide, Se substitution of the belt sulfides under turnover conditions has been exploited as an atom and site-specific spectroscopic marker to understand the local electronic structure, particularly the redox level, of the neighboring iron ions, via x-ray absorption spectroscopy (XAS).”

Reviewer #2 (Remarks to the Author):

Comment: Cutsail and coworkers synthesized and characterized a series of mixed-valent diiron dichalcogenide complexes, [L₂Fe₂Q₂] (Q = S, Se, Te), revealing that heavier chalcogenides (Se, Te) stabilize unprecedented S = 3/2 spin states due to reduced vibronic coupling. Using ⁵⁷Fe and ¹²⁵Te NRVS, they probed vibrational dynamics, demonstrating weaker Fe-Q coupling with increasing chalcogen mass. DFT calculations corroborated experimental findings, showing a shift of PKS normal modes to lower energies, leading to decreased vibronic coupling and stabilization of intermediate spin states. Their results highlight the influence of heavier chalcogens on electronic structure, with implications for designing spin-state-tuned materials. There are the following comments that need to be addressed:-

In the introduction, I suggest adding a few lines explaining why Se and Te were included in the study. While the discussion on sulfur-containing complexes is well-developed, the rationale for incorporating heavier chalcogenides is not explicitly mentioned. Clarifying this aspect would strengthen the motivation behind their inclusion.

Response: We believe there are multiple rationale for their inclusion, but perhaps one of the broadest appeals is the site-selective spectroscopic probes that the possible heavier chalcogen atoms offer. We have now added additional text:
“In addition to the direct study of Fe-S clusters, it has been recently demonstrated that heavier chalcogenides, both Se and Te, can be introduced in biological and biomimetic Fe-S clusters as additional spectroscopic probes, offering additional site-specificity.¹²⁻¹⁶”

Comment: The study emphasizes the role of heavier chalcogens in modulating spin states via vibrational effects, but what about the role of the β-diketiminato ligand scaffold? How does the steric or electronic tuning of these ligands impact exchange interactions and the extent of delocalization? Are there comparisons with other ligand frameworks to confirm the generality of the observed trends?

Response: Here we focused our investigation to the influence of the chalcogens. While we expect that the electronic and steric changes on the ligand backbone can have influence on the electron delocalization, particularly steric interactions and their effects on the Fe–Fe distances (and by extension the vibronic coupling), the complete examination of the ligand’s influence is beyond scope of this work. We are currently exploring this experimentally.

We note that a report from Driess and colleagues characterizing S and Se bridged clusters stabilized with less bulky nacnac ligands did not detail any S = 3/2 signals. See: <https://doi.org/10.1002/anie.201506788> and <https://doi.org/10.1002/ange.201706196>

Comment: The explanation of NRVS spectra trends is quite detailed, but the text could benefit from better structuring. The discussion jumps between experimental data and theoretical calculations without clear transitions. It would be useful to introduce a logical flow—first presenting experimental results comprehensively, then introducing the computational study, and finally comparing both.

Response: We appreciate the reviewer’s comment but find the suggestion to be a matter of preference. Considering the communication style of the manuscript and the journal, we prefer to write both results and discussion together. The structure we have offered though does generally separate the ⁵⁷Fe and ¹²⁵Te experiments, to allow for comparisons at the end. We have elected to keep the general outline of the paper unchanged but hope that the other revisions offered help to clarify this structure for the reviewer.

Comment: Some statements are repetitive, particularly when discussing peak shifts due to increasing chalcogen mass. The explanation of spectral shifts and in-plane vs. out-of-plane vibrations could be streamlined (I think the manuscript needs to be shortened a bit; it will benefit the readers).

Response: We agree with the reviewer and took the opportunity to very consciously reread and revised the manuscript to streamline it better. Various minor edits were made and kindly refer the reviewer to the manuscript with tracked changes.

Comment: The manuscript relies heavily on NRVS data to assign oxidation states and electronic structures. However, NRVS alone cannot definitively distinguish between localized and delocalized states. I recommend incorporating complementary techniques such as X-ray absorption spectroscopy (XAS) or Mössbauer spectroscopy to strengthen the conclusions regarding electronic localization and delocalization

Response: We did not apply NRVS to determine oxidation-state, or the degree of delocalization in these mixed-valent cluster, we fear there may be a confusion of the overall goal. All of these complexes were previously characterized by X-ray emission, EPR and Mössbauer spectroscopy along with magnetic susceptibility measurements (<https://doi.org/10.1038/s41557-021->

00853-5). All of the combined results clearly demonstrated the assigned and referenced spin-states we describe for complexes **1** - **4**. Adding these experiments again is wholly redundant.

From this comment we fear that that the main objective of this manuscript was overlooked by the reviewer; this study aims to directly measure the vibrational features of these compounds, and to evaluate the selectivity and differences between ^{125}Te and ^{57}Fe NRVS to resolve low-energy diiron dichalcogenide vibrations. The spectroscopic characterization of these low-energy vibrations is an essential component to understanding how these unique spin-states are stabilized – not what the spin-state is, as that is already known.

To this end, we have added more details concerning the experimental support of the spin-state assignment to the introduction.

Comment: Given the high-spin nature of these complexes, is single-determinant DFT sufficient for an accurate description?

Response: While intermediate spin states ($S = 3/2$) cannot be described by single-determinant DFT, we think that good agreement between experimental and calculated geometries, as well as between experimental and calculated NRVS spectra justifies our usage of DFT for vibrational analysis of the studied complexes. More advanced methods like CASSCF would require usage of a truncated model (smaller ligand) and would be performed on the DFT optimized (or partially optimized) geometry.

Table S1 shows that the chosen BP86 functional best reproduces the Fe₂Q₂ core structural parameters. Similar results were found by Benediktsson and Bjornsson (<https://doi.org/10.1021/acs.jctc.1c00753>) leading to a conclusion that functional with ~0 to ~15% HF exchange yield accurate FeS dimer structures. Though perhaps some caution is required before extending to very large clusters.

Additional text has been added to the manuscript on page 9 to further discuss this point.

“Although there are ab initio methods that demonstrate the ability to calculate intermediate spin-states they often rely on the truncated models and BS DFT geometry optimized structures.”

Comment: The manuscript suggests that complex **2** has a more localized electronic structure than complex **3**, but the claim lacks clear evidence, as the NRVS frequency shifts could be due to ligand field effects rather than electronic delocalization, and there is no supporting computational charge/spin density analysis; therefore, including spin density plots to directly visualize electron delocalization across the Fe centres would strengthen the argument.

Response: Unfortunately, we believe this comment comes again from a lack of familiarity with the proceeding work (<https://doi.org/10.1038/s41557-021-00853-5>). The Mössbauer spectrum of **2** exhibits Robin-Day Class II partial delocalization, while complex **3** is fully delocalized Class III. This has now been reiterated in the introduction through addition of the following sentences:

“In our case, ^{57}Fe Mössbauer experiments revealed two equally intense quadrupole doublets and isomer shift values for complexes **1** and **2**, consistent with a partially delocalized electronic structure (class II). Complex **3** on the other hand had only a single doublet, revealing a completely delocalized (class III) mixed valent ($\text{Fe}^{2.5+}\text{-Fe}^{2.5+}$) electronic structure.”

Additionally, we have added the calculated spin populations of the Fe and chalcogens to the SI (Table S2), which shows that total spin population on the chalcogens increase while spin on Fe decreases with introducing heavier chalcogens, further supporting increased delocalization in complex **3**.

Comment: I suggest that the author perform DFT optimizations considering multiple spin states, exploring all possible combinations of overall multiplicity. For example, in complex **1**, one Fe centre could be treated as an $S=2$ site, while the other could be considered an $S=3/2$ site, leading to overall multiplicities of either $S=1/2$ or $S=9/2$. Such an approach would allow the author to determine whether the molecule exhibits a ferromagnetically coupled ground state ($S=9/2$) or a lower-spin state ($S=1/2$). Additionally, I recommend providing the DFT-computed spin ladders, i.e., the relative energies of the $S=1/2$, $3/2$, $5/2$, $7/2$, and $9/2$ spin states for all complexes, along with visualizations of spin density distributions for each solution. These findings would offer valuable insights into the electronic structure and magnetic behavior of the mixed-valence system.

Response: We are hesitant to perform such calculations, as the formation of any locally intermediate spin iron sites has been experimentally excluded through Fe K-beta X-ray emission spectroscopy (<https://doi.org/10.1038/s41557-021-00853-5>). While it would be possible to perform the suggested calculations for $S(\text{tot}) = 3/2$, $5/2$ and $7/2$, those calculations would not represent the experimentally observed high spin Fe(II) and Fe(III) centers, and thus are not relevant to the complexes studied.

Furthermore, DFT spin ladders yield very inaccurate spin ladder energies. Ab initio methods such as DMRG more accurately predict spin ladder energies but these calculations are well beyond the scope of this paper and should be the focus of its own study.

The reviewer also suggests these calculations can distinguish between $S = 1/2$ and $9/2$ ground-state. Respectfully, this again exhibits a lack of familiarity with the work, as the ground-states are already known experimentally prior to this study and described in the introduction.

Comment: If the double exchange parameter (B) and magnetic exchange parameter (J) have already been reported, as cited in Figure 1, the author should explicitly state these values and further analyze the B/J ratio in relation to experimentally and computationally investigated vibronic coupling. It would be beneficial to include a separate discussion section that systematically highlights and compares these findings. Additionally, the author should clearly classify each molecule into mixed-valence Class I, II, or III, providing a rationale for these classifications. Furthermore, the discussion should emphasize the novel aspects of the study, particularly in the context of vibronic coupling, given that this concept has been previously explored.

Response: We appreciate the reviewer's suggestion and the values for J and B have been added and now first appear on page 16.

A section with discussion of influence of vibronic coupling on the electronic structure of studied complexes has been added, and studied complexes were assigned to appropriate mixed valence classes.

The concept of vibronic coupling leading to electronic localization is a well-established concept and our previous work only acknowledged and discussed its relevance as our novel findings contradicted common thinking. This manuscript now offers an experimental approach for the characterization of these low-energy vibrational modes utilizing nuclear resonance vibrational spectroscopy, and additional understanding of the role of heavy chalcogen substitution on the electronic structure. It is demonstrated that vibronic coupling not only influences electron delocalization but also has a significant influence on stabilization of intermediate spin states.

The new text is quoted below:

"... Although double exchange coupling has been estimated to be stronger than Heisenberg coupling in most $[\text{Fe}_2\text{S}_2]^+$ complexes,^{18,34} the combination of localizing contributions of Heisenberg exchange and vibrational coupling leads to partially delocalized $S = 1/2$ ground states (Robin-Day class II, Figure 1D). For antiferromagnetically coupled mixed-valent dimers that display complete electron delocalization ($S = 9/2$, Robin-Day class III),³⁵ it has been shown that the large double exchange interaction prevails over the localizing effects of Heisenberg exchange and vibrational couplings (Figure 1E).^{18,36,37} In our case, ^{57}Fe Mössbauer experiments revealed two equally intense quadrupole doublets and isomer shift values for complexes **1** and **2**, consistent with a partially delocalized electronic structure (class II).¹⁷ Complex **3** on the other hand had only a single doublet, revealing a completely delocalized (class III) mixed-valent ($\text{Fe}^{2.5+}\text{-Fe}^{2.5+}$) electronic structure.¹⁷"

"Calculated energy levels show that complex **1** displays a double-well $S = 1/2$ ground state in agreement with the previous partially delocalized class II assignment of the Robin-Day system (Figure 5A).¹⁷ Complex **2** can also be characterized as partially delocalized (class II, Figure 5) with $S = 1/2$ and $3/2$ states very close in energy, in agreement with experimentally observed mixture of these states.¹⁷ Unlike these, complex **3** shows fully delocalized $S = 3/2$ ground state (class III, Figure 5), which is also in line with experimentally observed intermediate $S = 3/2$ ground state.¹⁷"

Comment: The manuscript claims that chalcogen substitution ($\text{O} \rightarrow \text{S} \rightarrow \text{Se}$) significantly affects Fe oxidation states and electronic delocalization, but the justification is weak, as the only evidence provided—vibrational frequency shifts—may stem from mass effects rather than electronic changes, performing Natural Population Analysis (NPA) or Löwdin charge analysis, along with comparing experimental redox potentials, would provide stronger evidence.

Response: The reviewer (perhaps in a typo) has unfortunately stated the incorrect chalcogen series, as we do not study O and instead have heavily focused our work on (the omitted) Te.

Respectfully, we rebut the reviewer's assertion that the "justification is weak." As mentioned earlier, we make no claim that the NRVS evidences spin-state, or electronic delocalization. The experimental evidence for these characteristics was described in the introduction and cited. To further emphasize the already established spin-states, we have expanded the introduction, as detailed above.

More to the point of oxidation-state: previous Mössbauer characterization evidences the local oxidation-states and the degree of electronic delocalization. This point has been further clarified in the introduction/discussion.

Influence of chalcogen substitution on the electronic delocalization has been previously studied by Mössbauer spectroscopy (<https://doi.org/10.1038/s41557-021-00853-5>) which showed that complexes **1** and **2** are characterized by two overlapping quadrupole doublets (ratio 1:1) representing locally high spin Fe(II) and Fe(III) centers, while complex **3** is characterized by single quadrupole doublet with isomer shift value in between values for Fe(II) and Fe(III) centers in **1** and **2**. Based on Mössbauer studies it was determined that complexes **1** and **2** are partially localized mixed-valent clusters (Class II) while complex **3** is fully delocalized mix-valent cluster (Class III).

Lastly, the Mulliken spin populations are included in the SI (Table S2) as requested by the reviewer.

The following text is added to the introduction:

“In our case, ^{57}Fe Mössbauer experiments revealed two equally intense quadrupole doublets and isomer shift values for complexes **1** and **2**, consistent with a partially delocalized electronic structure (class II). Complex **3** on the other hand had only a single doublet, revealing a completely delocalized (class III) mixed valent ($\text{Fe}^{2.5+}\text{-Fe}^{2.5+}$) electronic structure.”

Comment: The authors assume that the highest-energy peaks correspond to Fe-Q vibrations but do not validate this with isotopic substitution, I don't know if it could be done or not.

Response: The highest energy transitions for complex **1** were attributed to Fe-S vibrations based on comparisons with assigned vibrations of previously characterized [2Fe-2S] clusters (<https://doi.org/10.1021/ja00192a002>, <https://doi.org/10.1021/ic0482584>, <https://doi.org/10.1021/bi701433m>, <https://doi.org/10.1021/acs.biochem.1c00252>).

In the case of complex **1**, the character of these vibrations was further supported by DFT calculations, which were also used to assign analogous vibrations for clusters **2** and **3**. For FeS clusters, isotopic substitution with ^{34}S has been performed in the past. However, such isotopic substitution experiments are beyond the scope of this work, as we are more interested in the low-energy vibrations.

Comment: DFT geometry optimizations were performed using the BP86 functional in ORCA. However, at the molecular level, the widely used hybrid B3LYP functional has been found to provide more accurate structural corrections. I suggest performing a few optimizations using the B3LYP functional and comparing the results with experimental structural parameters. This comparison would help clarify whether the GGA functional, such as BP86, is indeed superior to the hybrid functional in this case. Also, I would suggest that the author include vibrational frequencies along with corresponding oscillator strength to verify the global minimum of the optimised geometries.

Response: We performed our calculations with the following functionals: BP86, B3LYP, PBE, PBE0, TPSSh. We found that BP86 reproduces geometry of the diamond core better than other utilized functionals. Bond metrics, indicating this, are added to the SI (Table S1).

The calculated frequencies for the complexes are now included as supplemental information. The first submission had a single very small imaginary frequency (~ -25 - -10 cm^{-1}) for some of the complexes. We further optimized with a finer grid and no imaginary frequencies were found, supporting a global minimum. All spectra and frequencies are updated, and a maximum change of 1 cm^{-1} was observed. This does not impact any conclusions that were made previously.

Comment: In Figure 2., Indeed the author is able to explain and match the Fe-Q vibrational with the already reported {Fe-X} (X = Chalcogen) clusters. I would suggest providing a picture of all the vibrations (e.g. 139 cm^{-1} , 137 cm^{-1} , 112 cm^{-1} , 99 cm^{-1} , 283 cm^{-1} , etc.) showing the movement of atoms as given in Figures S11, S12 and S20. Interestingly, at lower energy vibrations, I can see a shoulder peak in molecule **1** (Figure 2) around 78 cm^{-1} , which downshifts in **2** to 75 cm^{-1} , and it disappears in the case of **3**. It would be interesting if the author explains about it.

Response: Vibrations at 139 cm^{-1} and 137 cm^{-1} in ^{57}Fe NRVS experimental spectra correspond to the DFT calculated vibrations at $127/128$ (**1**), $126/128$ (**2**), $124/126$ (**3**) cm^{-1} which are shown in SI (Figure S14). Vibration at 112 cm^{-1} in ^{57}Fe NRVS experimental spectrum of **1** correspond to the DFT calculated vibration at 101 cm^{-1} (Figure S13). Vibration at 99 cm^{-1} in ^{57}Fe NRVS experimental spectrum of **2** correspond to the DFT calculated vibration at 89 cm^{-1} (Figure S13). For vibration at 283 cm^{-1} in ^{57}Fe NRVS experimental spectrum of **1** it is not possible to assign it to a specific normal mode as intense PKS and in-plane Fe-S core vibrations are in this area.

The shoulder around 78 cm^{-1} can be attributed to the core in-plane vibration, mostly along y axis (DFT transition at 71 cm^{-1}). Analogous vibrations for complexes **2** and **3** are shifted as expected to lower energies, 51 cm^{-1} for **2** and 49 cm^{-1} for **3** based on DFT (see figure below). Discussion about this normal mode has not been included in the manuscript because this vibration cannot be unambiguously identified in the experimental spectra of complexes **2** and **3**. Feature around 75 cm^{-1} for complex **2** correspond to the different normal mode compared to the feature at 78 cm^{-1} for complex **1**, and based on DFT can be attributed to the twisting core in-plane vibration

Comment: Overall, I believe the manuscript in its current form doesn't meet the standard to be published in the Nat. Commun.

Response: We are disappointed that reviewer did not find our initial submission suitable for publication. In our revised manuscript, we have incorporated many of the reviewer's suggestions, while also expanding the introduction and discussion to further emphasize elements of the previous work that we believe the reviewer's suggestions would have had us duplicate.

Reviewer #3 (Remarks to the Author):

Comment: The manuscript presents a study of dinuclear mixed-valent iron complexes with S, Se, and Te bridges. All three have been previously reported by some of the authors, and the interesting property of the Se and Te clusters is that they exhibit intermediate total spin of $S = 3/2$ as opposed to the "normal" $S = 1/2$ low-spin situation (ref. 11). This arises due to the competition between localizing and delocalizing effects (vibronic coupling / double exchange). The manuscript approaches the subject using a combination of ^{57}Fe and ^{125}Te nuclear resonance vibrational spectroscopy (NRVS), a technique that can isolate vibrational modes associated with specific Mossbauer-active nuclei. In light of this work (as well as of ref. 46) the S/Te substitution as basis for NRVS, becomes a very important avenue for studying Fe/S systems in great local detail. The analysis is supported by computational models, which help in confirming the validity of the interpretations. For the present series of compounds the conclusions are quite clear, with a specific vibration (out of plane - "PKS") shown to become progressively dampened with heavier chalcogenides, allowing double exchange to dictate the intermediate spin state. Beyond the specific compounds, the study has exceptional value as an eloquent and compelling demonstration of how powerful this combined approach can be, with the Fe/Te-NRVS revealing distinct vibrations or enhancing different common vibrations in the low-energy part of the spectrum. The account of the research and the corresponding analysis are nothing less than a reference guide for how this approach is to be used and how chemically meaningful information can be extracted from such experiments.

Response: We are very pleased to read the reviewer's supportive comments, and we are also very happy that the reviewer clearly understood the aims and concept of our study. Like the reviewer suggests, we also hope that readers find our manuscript as a good demonstration of combining NRVS techniques to understanding complex phenomena and hopefully as guide for the novel ^{125}Te NRVS technique.

Reviewer #1 (Remarks to the Author):

The authors have adequately addressed my comments, and I am happy to support the revised manuscript for publication.

We thank the reviewer for their time and efforts in reviewing our manuscript and happy to read that we have their support.

Reviewer #2 (Remarks to the Author):

Cutsail and coworkers synthesized and characterized a series of mixed-valent diiron dichalcogenide complexes, $[L_2Fe_2Q_2]^-$ (Q = S, Se, Te), revealing that heavier chalcogenides (Se, Te) stabilize unprecedented $S = 3/2$ spin states due to reduced vibronic coupling. Using ^{57}Fe and ^{125}Te NRVS, they probed vibrational dynamics, demonstrating weaker Fe-Q coupling with increasing chalcogen mass. DFT calculations corroborated experimental findings, showing a shift of PKS normal modes to lower energies, leading to decreased vibronic coupling and stabilization of intermediate spin states. Their results highlight the influence of heavier chalcogens on electronic structure, with implications for designing spin-state-tuned materials. In the revised manuscript, the authors have attempted to address my previous comments; however, I find that some of the responses remain unconvincing or do not fully resolve the concerns I had raised—

We thank the reviewer for their time and efforts during the review process.

- (For comment 1) the response provided does not fully address my original concern. I believe the rationale for including Se and Te over S should be more clearly justified in terms of their impact on the electronic structure, rather than being implied through steric effects or ease of spectroscopic characterization. A brief explanation in the introduction highlighting the relevance of heavier chalcogens in tuning electronic properties—such as their ability to modulate metal–ligand covalency or influence spin state energetics—would significantly strengthen the scientific motivation for their inclusion in this study.

Our initial motivation for the synthesis of these clusters was for the introduction of an atomic specific probe; the consequence was perturbing the electronic structure. We find the general understanding of why and how the chalcogenide bridge perturbs the electronic structure lacking. We hope the initial spectroscopic focused goal clarifies our motivation.

Following sentence has been added to the Introduction:

Replacement of S with heavier chalcogenides have been used to tune properties of various systems¹⁷⁻¹⁹ including proteins²⁰⁻²², where they have impact on electronic structure, thus it is important to evaluate effect of chalcogen substitution on the electronic structure of biological and biomimetic clusters.

- While I understand the authors' preference for combining results and discussion, I believe that a clearer, section-wise comparison between the DFT and experimental findings—particularly in a side-by-side manner—would significantly enhance readability. Presenting the computational data alongside the corresponding experimental results within the Results section (rather than deferring comparisons to the general discussion) would allow the reader to more easily appreciate the alignment or discrepancies between the two. This structure would also reinforce the coherence of the NRVS spectral interpretation and avoid the current back-and-forth flow between theory and experiment. I still suggest reconsidering this presentation style to improve the manuscript's clarity and logical progression.

With all due respect, we have thoroughly considered and discussed the reviewer's suggestion, but we simply find ourselves at an impasse with the reviewer and will not be changing the order of the manuscript.

• I appreciate the authors' clarification regarding the primary objective of the manuscript and the prior characterization of the oxidation and spin states. However, even though these oxidation states have been thoroughly established in previous work using Mössbauer spectroscopy, X-ray emission, and EPR, I still believe it would benefit the reader if the manuscript included a concise sentence explicitly linking those earlier findings to the current study. A brief reference—perhaps in the introduction or results section—would help provide clearer context and reinforce how this vibrational study builds upon the previously established electronic structure. Including this one conclusive line would strengthen the narrative continuity for readers who may not be familiar with the prior work.

We are happy to read that we have now better presented our previous work. To draw a better line to the current work and explicitly describe how it builds upon the previous work, we have added the following text to the introduction:

“While previous EPR experiments revealed intermediate spin states and Mössbauer experiments established extent of electron delocalization, vibronic coupling and its influence was not directly studied.”

• While I acknowledge the authors' point that the ground states have been experimentally established in prior work, I still believe that incorporating DFT calculations for both the $S = 1/2$ and $S = 9/2$ states can offer valuable complementary insight—especially in the context of spin delocalization and magnetic exchange interactions. Since DFT studies on similar mixed-valent systems have already been reported, these calculations are feasible and can help visualize the spin density distribution across the metal centers. This, in turn, would clarify how ferro- and antiferromagnetic coupling contribute to the overall spin state. Therefore, I recommend that the authors report the relative DFT energies of the $S = 1/2$ and $S = 9/2$ states, considering both ferro- and antiferromagnetic alignments of the metal centers. Prior studies have shown that in mixed-valent systems, electronic oxidation or reduction can trigger a switch from antiferromagnetic to ferromagnetic coupling, and such a comparison here would be particularly relevant. Even if the absolute spin ladder energies from DFT are approximate, the qualitative trends can still provide meaningful insights that enrich the discussion of spin delocalization and electronic structure. (Dalton Trans., 2013, 42, 16490-16493, 10.1021/ja101887f, Phys. Chem. Chem. Phys., 2022, 24, 20760)

The manuscript includes calculation for both broken symmetry (BS) and high spin (HS) solutions. Figures S9 and S10 show ^{57}Fe and ^{125}Te NRVS for both BS and HS solutions. These figures are discussed in main text. We also expanded Table S2 to include spin populations and relative energies of HS states.

• In one of my earlier comments, what I intended to suggest was the following: select one of the most significant vibrational modes and use it to generate perturbed geometries along that mode. Then, compute the electronic structure for these displaced geometries to examine how the double exchange interaction varies with vibrational distortion. This approach would offer valuable insight into how specific vibrational motions influence double exchange and magnetic coupling. Even performing this analysis for just one representative molecule would allow for a meaningful magneto-structural correlation and provide a deeper understanding of the coupling between electronic and vibrational degrees of freedom. This is the core idea I was aiming to convey in my earlier suggestion.

We understand the reviewer's current suggestion but are unable to connect it with any other the previous comments in the first round. The new computational approach the reviewer is now suggesting is of significant interest, and we have already employed a significant amount of time and resources for such calculations (guided the approach seen here: <https://doi.org/10.1039/D3SC01402A>). However, we have found interesting disagreements with established protocols (specific to FeS clusters) that will require further investigation and optimization specifically for iron-chalcogenide clusters. We can confidently say that the calculations are beyond the scope of this manuscript.

The study addresses an important topic, and with the incorporation of the suggested minor revisions—particularly those clarifying the connection to prior work and enhancing the theoretical analysis—the authors could come back with a significantly strengthened manuscript suitable for reconsideration.

Reviewer #3 (Remarks to the Author):

The revised version offers important clarifications on certain points that might have been ambiguous or that assumed solid knowledge of prior literature (e.g. with respect to the electronic structure and oxidation states of the systems under study) or familiarity with specific technical/methodological aspects (e.g. with respect to the limits of DFT

applicability for intermediate spin states). In my view the authors sufficiently address or rebut all concerns raised on the initial submission. Understandably, requests that had to do with revisiting and replicating prior work cannot be addressed. I still view this work as an exemplary demonstration of combined Fe/Te NRVS and I expect that this study will become a reference point for a niche but powerful methodology.

We are very happy to read the reviewer's comments and appreciate their continued support of your study.